# The rhizobial effector NopT targets Nod factor receptors to regulate symbiosis in *Lotus japonicus*

Hanbin Bao[1†], Yanan Wang[1†], Haoxing Li[1], Qiang Wang[1], Yutao Lei[1], Ying Ye[1], Syed F Wadood[2], Hui Zhu[1], Christian Staehelin[2], Gary Stacey[3], Shutong Xu[1], Yangrong Cao[1*]

[1]National Key Lab of Agricultural Microbiology, Hubei Hongshan Laboratory, Huazhong Agricultural University, Wuhan, China; [2]State Key Laboratory of Biocontrol and Guangdong Key Laboratory of Plant Resources, School of Life Sciences, Sun Yat-sen University, Guangzhou, China; [3]Divisions of Plant Science and Technology, Christopher S. Bond Life Sciences Center, University of Missouri, Columbia, United States

**\*For correspondence:**
yrcao@mail.hzau.edu.cn

†These authors contributed equally to this work

**Competing interest:** The authors declare that no competing interests exist.

## eLife Assessment

This manuscript presents **important** findings on a bacterial effector involved in plant symbiotic signaling. The effector proteolytically targets a key receptor while its activity is counteracted by host-mediated phosphorylation, revealing a dynamic interplay that fine-tunes symbiotic interactions. The evidence supporting these claims is **solid**, and the findings have potential signaling implications beyond bacterial interactions with plants.

**Abstract** It is well documented that type-III effectors are required by Gram-negative pathogens to directly target different host cellular pathways to promote bacterial infection. However, in the context of legume–rhizobium symbiosis, the role of rhizobial effectors in regulating plant symbiotic pathways remains largely unexplored. Here, we show that NopT, a YopT-type cysteine protease of *Sinorhizobium fredii* NGR234 directly targets the plant's symbiotic signaling pathway by associating with two Nod factor receptors (NFR1 and NFR5 of *Lotus japonicus*). NopT inhibits cell death triggered by co-expression of NFR1/NFR5 in *Nicotiana benthamiana*. Full-length NopT physically interacts with NFR1 and NFR5. NopT proteolytically cleaves NFR5 both in vitro and in vivo, but can be inactivated by NFR1 as a result of phosphorylation. NopT plays an essential role in mediating rhizobial infection in *L. japonicus*. Autocleaved NopT retains the ability to cleave NFR5 but no longer interacts with NFR1. Interestingly, genomes of certain *Sinorhizobium* species only harbor *nopT* genes encoding truncated proteins without the autocleavage site. These results reveal an intricate interplay between rhizobia and legumes, in which a rhizobial effector protease targets NFR5 to suppress symbiotic signaling. NFR1 appears to counteract this process by phosphorylating the effector. This discovery highlights the role of a bacterial effector in regulating a signaling pathway in plants and opens up the perspective of developing kinase-interacting proteases to fine-tune cellular signaling processes in general.

## Introduction

The legume–rhizobium symbiosis stands as one of the most crucial and intricate mutualistic interactions in nature. Its significance goes beyond the supply of nitrogen fixed by rhizobia, which influences

plant growth and ecosystem sustainability, as this symbiosis is a pivotal model for investigating plant–microbe interactions (*Yang et al., 2022*). The orchestration of this symbiosis involves the perception of rhizobial signal molecules such as lipo-chitooligosaccharidic Nod factors (NFs) and surface polysaccharides (*Madsen et al., 2003*; *Tirichine et al., 2006*; *Kawaharada et al., 2015*), along with the secretion of proteins, including type three protein secretion system (T3SS) effectors translocated into host plants (*Sugawara et al., 2018*; *Zhang et al., 2021*).

Plant receptor heterocomplexes consisting of two Lysin-motif receptor kinases (LYKs) play a central role in recognizing various microbial poly- and oligosaccharide molecules. In non-legumes such as *Arabidopsis*, AtLYK5 and AtCERK1 (Chitin elicitor receptor kinase) are required for perception of chitin, and thereby trigger plant immunity to combat fungal pathogens (*Cao et al., 2014*). In legumes, structurally similar Nod factor receptors (NFRs) such as NFR1 and NFR5 in *Lotus japonicus* are vital for recognition of rhizobial NFs and initiation of symbiotic signaling (*Broghammer et al., 2012*; *Bozsoki et al., 2017*). Accordingly, *nfr1* and *nfr5* knockout mutants are almost completely unable to form nodules (*Madsen et al., 2003*; *Radutoiu et al., 2003*). Over-expression of either NFR1 or NFR5 can activate NF signaling, resulting in formation of spontaneous nodules in the absence of rhizobia (*Ried et al., 2014*). The intricate action of NFRs is highlighted by the elicitation of a hypersensitive cell death response when NFR1/NFR5 are simultaneously over-expressed in *Nicotiana benthamiana* leaves (*Madsen et al., 2011*). Similarly, defense-like responses in *Medicago truncatula* nodules were observed upon over-expression of MtNFP (NF perception, the ortholog of NFR5) (*Moling et al., 2014*), perhaps reflecting a potential role of MtNFP-interacting LYKs involved in plant–pathogen associations. It can therefore be expected that the protein abundance of NFRs must be tightly controlled to avoid resistance responses, thereby ensuring optimal root infection by rhizobia.

T3SS effectors of phytopathogenic Gram-negative bacteria play a critical role in suppressing pattern-triggered immunity (PTI) in host plants (*Jones and Dangl, 2006*). To avoid diseases, specific resistance (R) genes are employed to directly or indirectly recognize effectors translocated into host cells. Recognition of such avirulence (Avr) effectors often results in effector-triggered immunity (ETI), a much stronger immune response which is usually associated with programmed cell death (*Jones and Dangl, 2006*). In contrast to the well-studied effector functions of phytopathogenic bacteria in suppressing immunity through various mechanisms, it is largely unknown how T3SS effectors from symbiotic rhizobia contribute to mutualistic interactions.

T3SS effectors of the YopT-type family, such as YopT of *Yersinia pestis*, AvrPphB of *Pseudomonas syringae* pv. *phaseolicola* and rhizobial NopT proteins, are evolutionary related cysteine proteases (*Shao et al., 2002*; *Dai et al., 2008*). YopT cleaves Rho family GTPases leading to their inactivation to disrupt the actin cytoskeleton of human host cells (*Shao et al., 2003b*). AvrPphB can proteolytically cleave *Arabidopsis* receptor-like cytoplasmic kinase family proteins including PBS1, PBS1-like proteins and BIK1, to suppress PTI (*Shao et al., 2003a*; *Zhang et al., 2010*). Cleavage of PBS1 kinase by AvrPphB can be regarded as a 'decoy strategy' as it is associated with RPS5 (RESISTANCE TO PSEUDOMONAS SYRINGAE5)-mediated ETI (*Shao et al., 2003a*). Depending on the host legume, the NopT effector protease of *Sinorhizobium fredii* NGR234 can have an opposite function in regulating symbiosis (*Dai et al., 2008*; *Kambara et al., 2009*). The heterologous expression of NopT from NGR234 in *S. fredii* USDA257 severely blocks nodulation in soybean cultivar Nanfeng 15 but not in other soybean cultivars tested (*Khan et al., 2022*), implying that NopT might act as an Avr effector. In contrast, NopT positively regulates nodule formation in the interaction between *S. fredii* HH103 and soybean cultivar Suinong 14 (*Li et al., 2023*). NopT from *Bradyrhizobium* sp. ORS3257 promotes rhizobial infection in nodules of *Aeschynomene indica* (*Teulet et al., 2019*). NopT proteases have been biochemically characterized. Some exhibit autocleavage activity and are subsequently acylated at the newly formed N-terminus (*Dai et al., 2008*; *Dowen et al., 2009*; *Kambara et al., 2009*; *Fotiadis et al., 2012*; *Khan et al., 2022*). Recently, soybean PBS1-1 and two proteins in *Robinia pseudoacacia* were identified as NopT targets (*Luo et al., 2020*; *Khan et al., 2022*; *Li et al., 2023*). However, at the molecular level, it remains largely unknown how NopT modulates rhizobial infection and whether NopTs from different rhizobial strains differ in their effector activities.

A challenge in the investigation of rhizobial effectors lies in the availability of suitable assays and plants with which effector activities can be characterized. In this study, we took advantage of the observation that ectopic expression of NFR1 and NFR5 in *N. benthamiana* leaves induces programmed cell death (*Madsen et al., 2011*). We hypothesized that this response could be modulated by co-expressed

rhizobial effectors. Among the 15 known or putative T3SS effectors of *S. fredii* NGR234, only NopT could suppress NFR1/NFR5-induced cell death. This observation suggested that NopT is an effector associated with NFRs. Subsequent experiments demonstrated that NopT interacts with both NFR1 and NFR5 at the plasma membrane and proteolytically cleaves NFR5 at the juxtamembrane (JM) domain to suppress NF-mediated host responses. NFR1 phosphorylates the full-length NopT, thereby inactivating its protease activity. However, the autocleaved NopT form of *S. fredii* NGR234 retains the ability to cleave NFR5 but loses its ability to interact with NFR1. Intriguingly, various *Sinorhizobium* strains possess only truncated versions of NopT (related to autocleaved NopT of NGR234) which might evade the inactivation by NFR1. We present in this study a model of mutual regulation, in which the rhizobial effector NopT directly targets NFR5 to dampen symbiotic signaling, whereas NFR1-mediated phosphorylation of NopT counteracts NFR5 cleavage. Our work provides essential insights into the intricate interplay between legumes and rhizobia and shows an example of how a rhizobial effector fine-tune NF signaling by directly targeting NFRs, and how an NFR inactivates the enzymatic activity of the effector through phosphorylation.

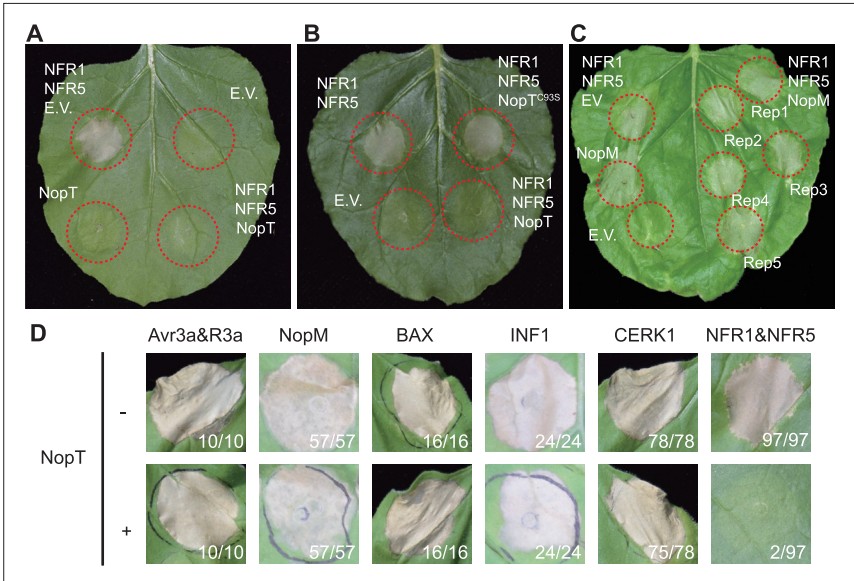

**Figure 1.** NopT specifically suppresses NFR1/NFR5-triggered cell death in *N. benthamiana*. *Agrobacterium* strains harboring plasmid DNA encoding NopT or the empty vector (EV) were infiltrated into *N. benthamiana* leaves. At 12 hpi, a second infiltration was performed with an *Agrobacterium* strain containing a plasmid with *NFR1/NFR5* genes. Dashed red lines indicate leaf discs where different proteins were expressed. NopT (**A**), but not NopT^C93S (**B**), a protease-inactive version of NopT, suppressed NFR1–NFR5-induced cell death. (**C**) Expression of NopM-induced cell death in *N. benthamiana*. (**D**) Expression of NopT could not suppress the cell death response triggered by expression of Avr3a/R3a, NopM, BAX, INF1, or AtCERK1. Numbers in (**D**) represent the number of leaf discs showing cell death and total leaf discs tested. Cell death in leaf disc results in the formation of necrotic plaques, which restrains pathogens within deceased cells. These plaques commonly manifest as leaf dehydration, frequently accompanied by a translucent appearance. Brown and shriveled leaf discs serve as indicators of cell death. The pictures shown in this figure are representative of at least three independent biological replicates.

The online version of this article includes the following source data and figure supplement(s) for figure 1:

**Figure supplement 1.** Analyses of the functions of all 15 effectors from *S. fredii* NGR234 in preventing NFR1- and NFR5-induced cell death in *N. benthamiana* leaves.

**Figure supplement 1—source data 1.** Original files for western blot analysis displayed in *Figure 1—figure supplement 1E*.

**Figure supplement 1—source data 2.** PDF file containing original western blots for *Figure 1—figure supplement 1E*, indicating the relevant bands and treatments.

## Results

### NopT expression in *N. benthamiana* suppresses the NFR1/NFR5-induced cell death response

To examine the role of effector proteins in regulating rhizobial infection, 15 effector genes (***Kimbrel et al., 2013***) were cloned from *S. fredii* NGR234, a strain with an exceptionally broad host range. Initial experiments were performed to screen the effector proteins for their ability to suppress or enhance the NFR1/NFR5-triggered cell death response of *N. benthamiana* leaves. In this experiment, each effector was co-expressed with NFR1/NFR5 (***Figure 1—figure supplement 1A–C***). Only NopT, a cysteine protease belonging to the YopT family (***Shao et al., 2002***; ***Dai et al., 2008***), could suppress NFR1 and NFR5-induced cell death (***Figure 1A***, ***Figure 1—figure supplement 1D–F***). Interestingly, NopT reduced protein abundance of the intact NFR5 when co-expressed with NFR1 (***Figure 1—figure supplement 1E***), suggesting that NopT might promote proteolytic degradation of NFR5. Cys-93 of NopT is an amino acid residue known to be essential for the protease activity of this effector (***Dai et al., 2008***; ***Kambara et al., 2009***). In contrast to the wild-type NopT protein, expression of the protease-dead NopT$^{C93S}$ form in *N. benthamiana* did not result in suppression of cell death induced by NFR1/NFR5 (***Figure 1B***), indicating that the protease activity of NopT is essential for the observed cell death-suppressing activity of NopT. In contrast to NopT, NopM, an effector protein from *S. fredii* NGR234 reported to possess E3 ligase activity (***Xin et al., 2012***), induced cell death when expressed in *N. benthamiana* leaves under our test conditions (***Figure 1C***).

We also tested whether the cell death-suppressing effect of NopT was specific for NFR1/NFR5 expression or whether NopT could act generally to suppress cell death. In addition to NopM, we

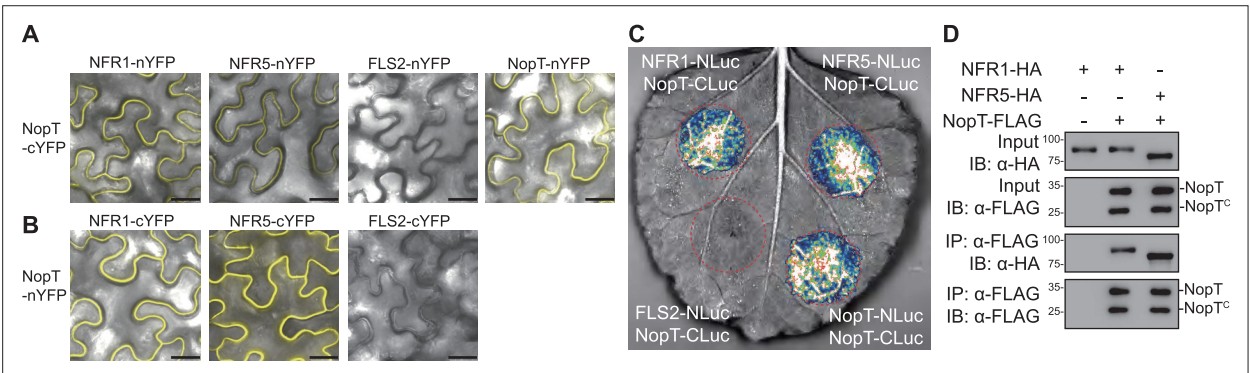

**Figure 2.** NopT interacts with NFR1 and NFR5. Interactions between NopT and NFR1 or NFR5 were detected using bimolecular fluorescence complementation (BiFC) (Split-YFP) (**A, B**), Split-LUC complementation (**C**) and co-IP (**D**) assays in *N. benthamiana* leaves. (**A, B**) For BIFC analysis, nYFP and cYFP tags were fused at the C-terminus of NopT, NFR1, NFR5, and the flagellin receptor FLS2 (negative control). YFP fluorescence signals represent protein–protein interactions. Scale bar = 25 μm. (**C**) For the Split-LUC complementation assay, the NLuc and CLuc tags were fused at the C-terminus of NopT, NFR1, NFR5, and FLS2 (negative control). Luminescence signals represent protein–protein interactions. (**D**) For the co-IP assay, HA-tagged NFR1 or NFR5 and FLAG-tagged NopT were expressed in *N. benthamiana* cells followed by immunoprecipitation using an anti-FLAG antibody. Immunoblot analysis was performed using anti-HA and anti-FLAG antibodies (NopT$^C$ denotes the truncated version of NopT after autocleavage). The images and immunoblots shown in this figure are representative of three biological replicates.

The online version of this article includes the following source data and figure supplement(s) for figure 2:

**Source data 1.** Original files for western blot analysis displayed in ***Figure 2D***.

**Source data 2.** PDF file containing original western blots for ***Figure 2D***, indicating the relevant bands and treatments.

**Figure supplement 1.** NopT interacts with both NFR1 and NFR5.

**Figure supplement 1—source data 1.** Original files for western blot analysis displayed in ***Figure 2—figure supplement 1***.

**Figure supplement 1—source data 2.** PDF file containing original western blots for ***Figure 2—figure supplement 1***, indicating the relevant bands and treatments.

**Figure supplement 2.** Split-LUC assay testing the interactions between different NopT mutants and NFRs.

**Figure supplement 2—source data 1.** Original files for western blot analysis displayed in ***Figure 2—figure supplement 2B***.

**Figure supplement 2—source data 2.** PDF file containing original western blots for ***Figure 2—figure supplement 2B***, indicating the relevant bands and treatments.

examined various proteins known to induce cell death when expressed in *N. benthamiana* leaves, namely Avr3a and R3a (*Bos et al., 2006*), apoptosis regulator BAX (Bcl-2-associated X protein) (*Gan et al., 2009*), INF1, an elicitor from *Phytophthora infestans* (*Vleeshouwers et al., 2006*) and *Arabidopsis* AtCERK1 (*Cao et al., 2014*; *Figure 1C*). In contrast to the NFR1/NFR5-induced cell death, expression of NopT was unable to suppress cell death when each of these proteins was expressed in *N. benthamiana* leaves (*Figure 1D*). Hence, the effect of NopT appears to be specific for the NFRs. These results suggested that the effector protease NopT might act directly on NFR1 and/or NFR5.

## NopT interacts with NFR1 and NFR5

Autocleaved NopT of *S. fredii* NGR234 contains acylation sites required for lipidation and subsequent plasma membrane localization (*Dowen et al., 2009*; *Khan et al., 2022*). To investigate whether NopT interacts with NFR1 and NFR5, we first used in vivo bimolecular fluorescence complementation (BiFC) analysis. NopT, NFR1, and NFR5 were C-terminally tagged with nYFP and cYFP. Fluorescence signals representing direct NopT–NFR1, NopT–NFR5, and NopT–NopT interactions were detected at the plasma membrane of *N. benthamiana* (*Figure 2A, B*). Interactions between NopT and NFR1 and NFR5 were further verified using a Split-luciferase (Split-LUC) complementation assay (*Figure 2C*). Co-expression of NopT-CLuc and NFR1-NLuc or NFR5-NLuc produced strong luminescence signals in transformed leaf discs compared with the negative control expressing AtFLS2-NLuc (*Arabidopsis* FLAGELLINSENSING 2) and NopT- CLuc (*Figure 2C*).

It is known that acylation of NopT can alter its subcellular localization (*Dowen et al., 2009*; *Khan et al., 2022*). Here, we performed Split-LUC assays to examine the interaction between NFR1/NFR5 and NopT$^{G50A/C51A/C52A}$, a modified NopT form lacking acylation sites. Compared to the wild-type form (NopT$^{WT}$), the interactions between NopT$^{G50A/C51A/C52A}$ and NFR1/NFR5 were significantly reduced in Split-LUC assays (*Figure 2—figure supplement 2A, B*). However, the protease-dead version of NopT (NopT$^{C93S}$) showed interactions with NFR1 and NFR5, which were different from NopT$^{WT}$ (*Figure 2—figure supplement 2C*). These results were further confirmed by a co-immunoprecipitation (co-IP) experiment in *N. benthamiana* leaves co-expressing FLAG-tagged NopT with C-terminally HA-tagged NFR1 or NFR5. Both NFR1 and NFR5 could be co-precipitated by an anti-FLAG antibody when the sample contained NopT-FLAG (*Figure 2D*, *Figure 2—figure supplement 1*). Overall, the identified physical interactions between NopT and NFR1/NFR5 indicated that NopT targets NFRs and that NFR5 and/or NFR1 may be proteolytically cleaved by NopT.

## NopT proteolytically cleaves NFR5 at the JM domain

A previous study has shown that NopT is autocleaved at its N-terminus to form a processed protein that lacks the first 49 amino acid residues (*Dai et al., 2008*). To test whether NopT could proteolyze NFR1 and/or NFR5, NopT was co-expressed with NFR1-GFP and NFR5-GFP fusion proteins in *N. benthamiana* leaves. The expressed proteins were separated by SDS–PAGE and subjected to immunoblotting. Proteolytic cleavage of NFR5-GFP but not NFR1-GFP was observed in the presence of co-expressed NopT, whereas protease-dead NopT$^{C93S}$ showed no effect (*Figure 3A*, *Figure 3—figure supplement 1A*). A similar in vivo cleavage assay with transgenic *L. japonicus* roots also showed that expression of NopT but not NopT$^{C93S}$ caused proteolytic cleavage of NFR5 (*Figure 3B*). Based on the molecular weight changes, the cleavage site in NFR5 was predicted to occur within the cytoplasmic domain (CD) of NFR5. To further explore NFR5 cleavage by NopT, the CD of NFR5 (Strep-NFR5$^{CD}$-HA) and NopT-FLAG or protease-dead NopT$^{C93S}$-FLAG were co-expressed in the *Escherichia coli* cells. Immunoblotting results showed that the active NopT protease cleaved NFR5$^{CD}$ resulting in a ~5-kDa smaller protein (*Figure 3C, D*). However, when the CD of NFR1 was co-expressed with NopT in the same experiment, no NFR1 cleavage was detected (*Figure 3C*). Migration of the bands representing full-length NopT or NopT$^{C93S}$ was significantly retarded on the gel when the CD of NFR1 was co-expressed (*Figure 3C*), suggesting that NopT might be phosphorylated by NFR1 in *E. coli* cells.

The NFR5$^{CD}$ cleavage product with a ~5-kDa lower molecular weight in the cleavage assay suggested that the cleavage site is located within the JM domain of NFR5. To test this hypothesis, we replaced the JM with the SUMO tag to create SUMO-NFR5$^{KC}$-HA (KC, kinase domain and C-terminal tail region, a modified NFR5$^{CD}$ without the JM) for proteolytic assay. Indeed, immunoblot analysis showed that co-expressed NopT was unable to cleave SUMO-NFR5$^{KC}$-HA (*Figure 3—figure supplement 1B*). The JM domain of NFR5 was then fused to SUMO and GFP to create a recombinant

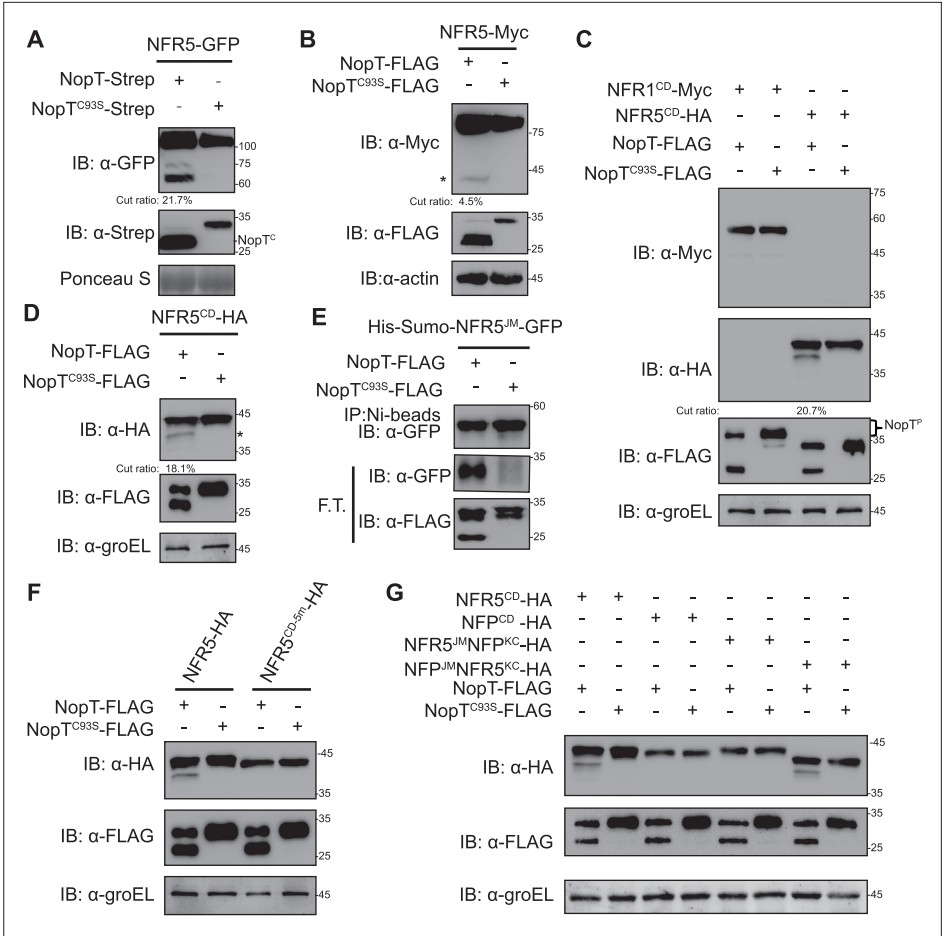

**Figure 3.** NopT proteolyzes NFR5 at its juxtamembrane (JM) domain. Proteins with indicated tags were expressed in *N. benthamiana* (**A**), *L. japonicus* (**B**), or *E. coli* cells (**C–G**) and detected by immunoblotting. (**A**) NopT but not NopT[C93S] (a protease-dead version of NopT) cleaves NFR5-GFP protein expressed in *N. benthamiana* cells by releasing NFR5[CD]-GFP (NopT[C] denotes autocleaved NopT). The cleavage efficiency was marked under the lane. (**B**) NopT but not NopT[C93S] cleaves NFR5-Myc expressed in hairy roots of *L. japonicus* by releasing NFR5[CD]-Myc. The cleavage efficiency was marked under the lane. The asterisk indicates the HA-tagged NFR5 cleavage product containing the kinase domain and a C-terminal tail region. (**C**) Analysis of the CDs of NFR1 and NFR5 co-expressed with NopT or NopT[C93S] in *E. coli* cells. Cleavage of NFR5[CD] was observed for NopT but not NopT[C93S], while NFR1[CD] was not proteolyzed by NopT. In the presence of NFR1[CD], a slower migrating band was observed, possibly representing phosphorylated NopT (NopT[P]). The cleavage efficiency was marked under the lane. (**D**) A repeat experiment confirmed that NopT is able to cleave NFR5[CD] (the asterisk indicates the HA-tagged NFR5 cleavage product containing the kinase domain and a C-terminal tail region). The cleavage efficiency was marked under the lane. (**E**) NopT cleaves His-SUMO-NFR5[JM]-GFP (His-SUMO and GFP linked by the juxtamembrane [JM] domain of NFR5) in vitro. F.T. indicates proteins in flow through samples after purification with Ni-beads. (**F**) NopT expressed in *E. coli* cells was unable to cleave co-expressed NFR5[CD-5m]-HA, a modified version of NFR5[CD]-HA in which five amino acids of the JM were substituted by other residues (S283Y, G294Q, Y303S, A310I, and T311Y). (**G**) NopT expressed in *E. coli* cells was unable to cleave co-expressed *M. truncatula* NFP[CD]-HA and the NFR5[JM]-NFP[KC]-HA fusion protein, while NFP[JM]-NFR5[KC] was proteolyzed (KC stands for the kinase domain and a C-terminal tail region, modified Nod factor receptors without the JM).

The online version of this article includes the following source data and figure supplement(s) for figure 3:

**Source data 1.** Original files for western blot analysis displayed in *Figure 3A–G*.

**Source data 2.** PDF file containing original western blots for *Figure 3A–G*, indicating the relevant bands and treatments.

**Figure supplement 1.** NopT cleaves NFR5 but not NFR1.

**Figure supplement 1—source data 1.** Original files for western blot analysis displayed in *Figure 3—figure*

*Figure 3 continued on next page*

*Figure 3 continued*

**supplement 1A and B**.

**Figure supplement 1—source data 2.** PDF file containing original western blots for *Figure 3—figure supplement 1A and B*, indicating the relevant bands and treatments.

**Figure supplement 2.** NopT interacts with the juxtamembrane (JM) domain of NFR5.

**Figure supplement 2—source data 1.** Original files for western blot analysis displayed in *Figure 3—figure supplement 2*.

**Figure supplement 2—source data 2.** PDF file containing original western blots for *Figure 3—figure supplement 2*, indicating the relevant bands and treatments.

**Figure supplement 3.** Conserved domains and residues of NFR5 and related proteins.

**Figure supplement 4.** NopT cleaves NFR5 at the juxtamembrane domain.

**Figure supplement 4—source data 1.** Original files for western blot analysis displayed in *Figure 3—figure supplement 4A–G*.

**Figure supplement 4—source data 2.** PDF file containing original western blots for *Figure 3—figure supplement 4A–G*, indicating the relevant bands and treatments.

**Figure supplement 5.** Mass spectrometry analysis of cleavage site of NFR5 by NopT.

**Figure supplement 6.** NopT cleaves the NFR5 homolog proteins at the juxtamembrane (JM) domain.

**Figure supplement 6—source data 1.** Original files for western blot analysis displayed in *Figure 3—figure supplement 6*.

**Figure supplement 6—source data 2.** PDF file containing original western blots for *Figure 3—figure supplement 6*, indicating the relevant bands and treatments.

**Figure supplement 7.** NopT cleaves the juxtamembrane (JM) of MtNFP.

**Figure supplement 7—source data 1.** Original files for western blot analysis displayed in *Figure 3—figure supplement 7*.

**Figure supplement 7—source data 2.** PDF file containing original western blots for *Figure 3—figure supplement 7*, indicating the relevant bands and treatments.

**Figure supplement 8.** NopT cleaves recombinant proteins NFR5$^{268-445}$-NFP$^{458-595}$ and NFP$^{270-457}$-NFR5$^{456-595}$.

**Figure supplement 8—source data 1.** Original files for western blot analysis displayed in *Figure 3—figure supplement 8*.

**Figure supplement 8—source data 2.** PDF file containing original western blots for *Figure 3—figure supplement 8*, indicating the relevant bands and treatments.

---

protein, SUMO-NFR5$^{JM}$-GFP. In an in vitro cleavage assay with recombinant proteins from *E. coli* cells, GFP was immunodetected in the flow though sample when SUMO-NFR5$^{JM}$-GFP was incubated with NopT but not with NopT$^{C93S}$ (*Figure 3E*). The interaction between NopT$^{C93S}$ and SUMO-NFR5$^{JM}$-GFP was confirmed by an in vitro pull-down assay (*Figure 3—figure supplement 2*). Overall, these data strongly indicate that the cleavage site of NFR5 for NopT is located in its JM domain.

## The cleavage of NFR5 by NopT is dependent on multiple residues of the JM domain

The 'DKLLSGV' motif in the JM domain of NFR5 (residues Asp-288 to Val-294; *Figure 3—figure supplement 3*) shows highest similarity with the autocleavage region of NopT (DKMGCCA). However, a protein variant of NFR5$^{CD}$ in which the 'DKLLSGV' motif was replaced by seven alanine residues could still be cleaved by NopT (*Figure 3—figure supplement 4A*). Similar to NopT, the AvrPphB effector of *P. syringa* has an autocleavage site (*Shao et al., 2003a*). We therefore also examined a version of NFR5$^{CD}$ in which the 'DKLLSGV' motif was replaced by that of AvrPphB. In the cleavage assay with *E. coli* cells, AvrPphB did not cleave NFR5$^{CD}$. However, the modified NFR5$^{CD}$ form with the seven residues from AvrPphB was cleaved by AvrPphB and the cleavage product had a lower molecular weight than that formed by NopT (*Figure 3—figure supplement 4A*). These findings indicated that the cleavage site for NopT is located upstream of the 'DKLLSGV' motif in NFR5.

As mentioned above, NopT undergoes autocleavage but it is not clear whether this is due to inter- or intramolecular proteolysis. In addition to Cys-93, His-205, and Aps-220 of NopT are conserved

catalytic residues (*Dai et al., 2008*; *Kambara et al., 2009*). Therefore, NopT forms lacking these catalytic residues were co-expressed with NopT in *E. coli* cells. The immunoblot analysis showed that NopT could not proteolyze NopT$^{C93S}$, NopT$^{H205A}$, or NopT$^{D220A}$ (*Figure 3—figure supplement 4B*), suggesting that NopT autocleavage is due to intramolecular proteolysis. Hence, the residues at the autocleavage site of NopT may not help to predict the cleavage site of NopT substrates.

In an attempt to define the NopT cleavage site in NFR5, we created a series of modified NFR5 forms covering the JM domain of NFR5 (Val-269 to Cys-320) and the first seven amino acid residues of the kinase domain (Lys-321 to Tyr-327). Initially, 17 variants of NFR5$^{CD}$ were created in which three adjacent amino acids in the JM domain were replaced with three alanine residues. When these proteins were co-expressed with NopT in *E. coli* cells, all 17 NFR5$^{CD}$ variants could be cleaved by NopT but not NopT$^{C93S}$ (*Figure 3—figure supplement 4C–F*). In a similar experiment, we created seven NFR5$^{CD}$ forms which contained a deletion in the JM domain of 10 residues each. Surprisingly, cleavage products were still observed for all seven examined proteins (*Figure 3—figure supplement 4G*). These results suggest that the JM domain may have multiple sites that can be cleaved by NopT, which is different from the specific cleavage site identified for AvrPphB. In another experiment, recombinant His-SUMO-NFR5$^{CD}$-GST protein was co-expressed with NopT in *E. coli* cells and its cleavage product was subjected to N-terminal protein sequencing using liquid chromatography–mass spectrometry (LC–MS) analysis. Based on alignment of identified peptides, the NopT cleavage site in NFR5 was mapped to four basic amino acids in the JM domain (RRKK; amino acid residues 271–275) (*Figure 3—figure supplement 5*). However, this result was not consistent with the observation that NopT expressed in *E. coli* cells could still proteolyze a co-expressed NFR5$^{CD}$ form in which the residues from Tyr-268 to Leu-277 were deleted (*Figure 3—figure supplement 4G*). Overall, the experiments performed suggest that NopT preferentially proteolyzes NFR5 at the RRKK motif, but other sites in the JM domain can also be cleaved by NopT. Since NFR5 belongs to a subfamily of LYK family proteins that lacks kinase activity (*Yang et al., 2022*), we wondered whether NopT could proteolyze other LYK proteins of this subfamily, which might be helpful to characterize the biochemical function of NopT. We therefore investigated whether the CDs from *Arabidopsis* AtLYK5, *L. japonicus* LjLYS11, and *M. truncatula* MtNFP expressed in *E. coli* cells can be proteolyzed by NopT. As shown in *Figure 3—figure supplement 6*, NopT, but not NopT$^{C93S}$, was able to cleave the CDs of AtLYK5 and LjLYS11. Replacement of the JM domain in NFR5$^{CD}$ with the JM domain from either AtLYK5 or LjLYS11 resulted in cleavage by NopT (*Figure 3—figure supplement 6*). These data show that NopT cleaves AtLYK5 and LjLYS11 at their JM domains. Based on sequence analysis of the JM region of NFR5 and its homologous proteins, five conserved residues (Ser-283, Gly-294, Tyr-304, Ala-310, and Thr-311) were identified and mutated as NFR5$^{CD-5m}$ for the proteolytic assay (*Figure 3—figure supplement 3B*). As shown in *Figure 3F*, NFR5$^{CD-5m}$, a NFR5$^{CD}$ variant with point mutations of S283Y, G294Q, Y304S, A310I, and T311Y, could not be cleaved by NopT. Interestingly, NopT could not proteolyze MtNFP$^{CD}$ (*Figure 3G*). We then swapped the JM domains in NFR5$^{CD}$ and MtNFP$^{CD}$, generating NFP$^{JM}$-NFR5$^{KC}$ and NFR5$^{JM}$-NFP$^{KC}$ (KC stands for the kinase domain and the C-terminal tail region). As shown in *Figure 3G*, NopT could proteolyze NFP$^{JM}$-NFR5$^{KC}$ but not NFR5$^{JM}$-NFP$^{KC}$ in *E. coli* cells. In an in vitro cleavage assay, the cleavage product of SUMO-NFP$^{JM}$-GFP proteolyzed by NopT was detected in the flow though sample (*Figure 3—figure supplement 7*). These data suggest that the cleavage of NFR5 by NopT depends on the conformational structure of NFR5$^{KC}$ and/or other specific amino acids in NFR5$^{KC}$. We then investigated whether two additional recombinant proteins with large swapping regions in NFR5$^{CD}$ and NFP$^{CD}$ (NFR5$^{268-445}$-NFP$^{458-595}$ and NFP$^{270-457}$-NFR5$^{456-595}$) could be cleaved in the *E. coli* proteolysis test (*Figure 3—figure supplement 8*). Notably, both proteins were cleaved by NopT, suggesting that a specific conformation of the NopT substrate is required for proteolysis. Taken together, we concluded from our experiments that NopT can proteolyze NFR5 at the JM domain and that multiple nonadjacent residues in NFR5$^{JM}$ in combination with a specific KC conformation are required for proteolysis.

## NopT phosphorylated by NFR1 is proteolytically inactive

The perception of rhizobial NFs by the heterocomplex of NFR1 and NFR5 involves trans-phosphorylation events from the NFR1 kinase to the pseudo-kinase NFR5 (*Madsen et al., 2011*). Since NopT is associated with both NFR1 and NFR5, we hypothesized that NopT is a phosphorylation target of NFR1, which may interfere with transphosphorylation events between the two receptors. NopT$^{C93S}$ was therefore expressed in *L. japonicus* wild-type and *NFR1* knockout mutant plants (*nfr1-1*) and analyzed

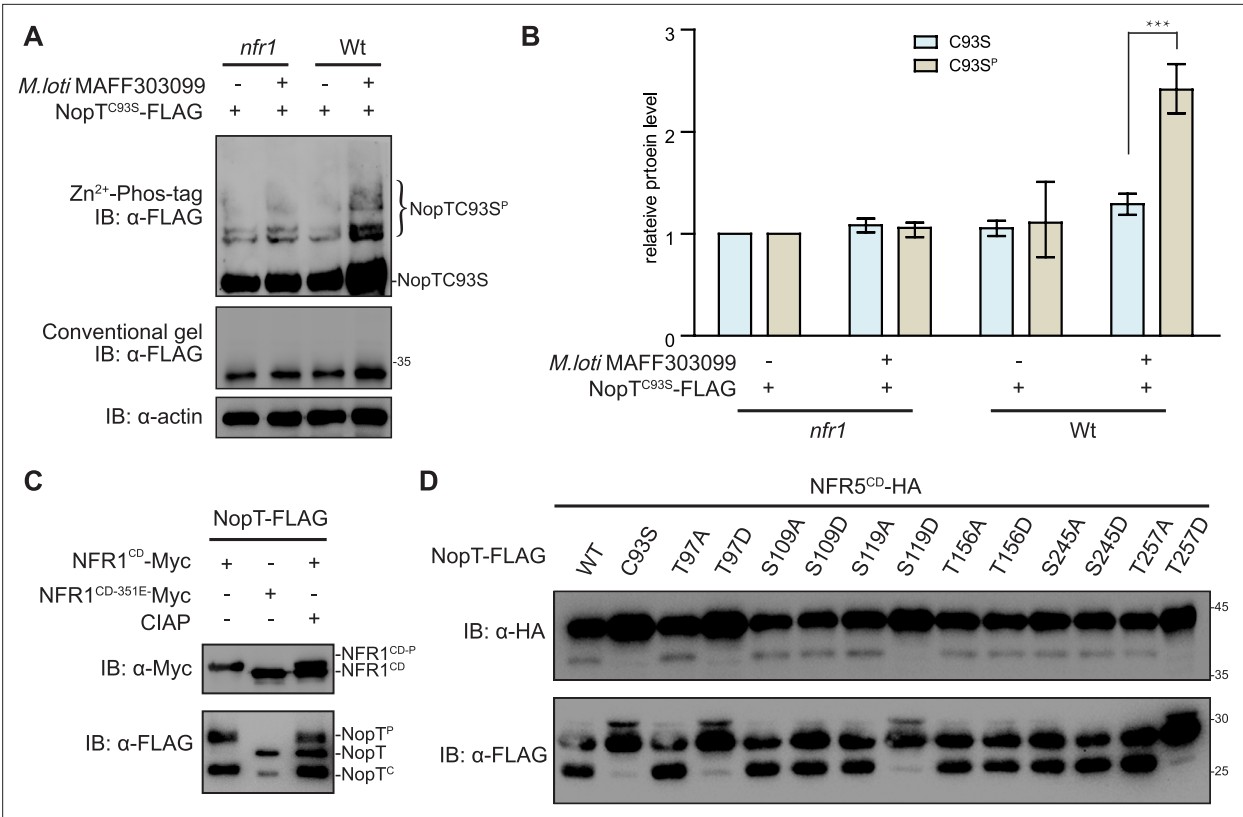

**Figure 4.** Phosphorylation of NopT by NFR1 suppresses its proteolytic activity. (**A**) In vivo phosphorylation assay with proteins expressed in *L. japonicus* roots (wild-type and *nfr1-1* mutant plants) using $Zn^{2+}$-Phos-tag SDS–PAGE. Phosphorylation of NopT[C93S] was induced by inoculation with rhizobia (*Mesorhizobium loti* MAFF303099) and was largely dependent on NFR1. (**B**) The relative protein amount of each lane, as shown in (**A**), was quantified with ImageJ software (three biological replicates, student's *t*-test; *** indicates statistically significant differences at p<0.01). The value of the control band in each gel was set to 1 for comparison. Values are means ± SEM. (**C**) NFR1[CD] but not NFR1[CD-K351E] phosphorylates NopT in *E. coli* cells. The phosphorylated full-length form of NopT could be dephosphorylated by calf intestinal alkaline phosphatase (CIAP) (gel band shift). Abbreviations: NFR1[CD-P], autophosphorylated NFR1; NopT[P], phosphorylated NopT; NopT, nonphosphorylated NopT; NopT[C], autocleaved NopT. (**D**) The phosphorylation sites of NopT identified by liquid chromatography–mass spectrometry (LC–MS) were either substituted to alanine (**A**) or aspartate (**D**). The indicated NopT variants were subsequently tested for autocleavage and NFR5[CD] proteolysis in *E. coli* cells. Wild-type NopT (WT) and protease-dead NopT[C93S] were included into the analysis.

The online version of this article includes the following source data and figure supplement(s) for figure 4:

**Source data 1.** Original files for western blot analysis displayed in *Figure 4A, C and D*.

**Source data 2.** PDF file containing original western blots for *Figure 4A, C and D*, indicating the relevant bands and treatments.

**Figure supplement 1.** NFR1[CD] phosphorylates NFR5[CD] in vitro.

**Figure supplement 1—source data 1.** Original files for western blot analysis displayed in *Figure 4—figure supplement 1*.

**Figure supplement 1—source data 2.** PDF file containing original western blots for *Figure 4—figure supplement 1*, indicating the relevant bands and treatments.

on $Zn^{2+}$-Phos-tag gels. On such gels, a $Zn^{2+}$-Phos-tag-bound phosphorylated protein migrates slower than its unbound nonphosphorylated form. Slower migrating bands corresponding to phosphorylated NopT[C93S] were observed, particularly when wild-type plants were inoculated with rhizobia (*Figure 4A, B*). These findings indicated that NopT phosphorylation in planta is largely dependent on NFR1 and that the protein was probably phosphorylated at multiple sites.

Next, we performed assays to investigate whether expressed NFR1[CD] directly phosphorylates co-expressed NopT. NFR5[CD] and a kinase-inactive NFR1[CD] form (NFR1[CD-K351E]; Lys-351 is a conserved residue required for ATP binding) were included into these experiments. When NFR1[CD] was expressed in *E. coli* cells, extracted NopT (*Figures 3C and 4C*) and NFR5[CD] (*Figure 4—figure supplement 1*) exhibited retarded migration on SDS–PAGE gels, suggesting that NFR1 was able to phosphorylate both proteins. A band shift was observed when the phosphorylated full-length NopT was in vitro

dephosphorylated by calf intestinal alkaline phosphatase (CIAP). However, the CIAP treatment caused no obvious band shift of the autocleaved NopT form (*Figure 4C*). The phosphorylation sites of NopT were then identified using an in vitro phosphorylation assay followed by LC–MS analysis. Several serine and threonine residues in NopT were identified to be phosphorylated by NFR1$^{CD}$ (*Supplementary file 1a*). Taken together, these results showed that full-length NopT is a phosphorylation target of the NFR1 kinase.

Finally, we wondered whether NopT phosphorylation by NFR1 influences its proteolytic activity. Plasmids encoding NopT variants were constructed in which the phosphorylated residues were substituted to either alanine (to block phosphorylation) or aspartate (to mimic the phosphorylation status). The different NopT forms were then expressed in *E. coli* cells to analyze NopT autocleavage and proteolysis of co-expressed NFR5$^{CD}$. Aspartate substitutions at three phosphorylation sites of NopT (Thr-97, Ser-119, and Thr-257) resulted in the loss of the autoproteolytic activity, as well as the ability to cleave NFR5$^{CD}$ (*Figure 4D*). In contrast, corresponding NopT proteins with alanine substitutions retained autocleavage activity and the ability to proteolyze NFR5$^{CD}$ (*Figure 4D*). These data indicate that phosphorylation of full-length NopT by NFR1 inhibits its proteolytic activity, thereby keeping NopT unprocessed and NFR5 uncleaved.

## NopT dampens rhizobial infection

As NopT of *S. fredii* NGR234 could directly target and cleave NFR5, we expected that NopT may be an important player regulating rhizobial infection in legumes. NF signaling is known to function during early stages of rhizobial infection (*Geurts et al., 2005*; *Wang et al., 2012*; *Cai et al., 2018*). We focused our experiments on early stages of rhizobial infection. Inoculation of *L. japonicus* plants with a GFP- or GUS-labeled *nopT* mutant of NGR234 (NGR234Δ*nopT*) resulted in a massive infection of root hairs, whereas infection by the NGR234 wild-type strain was less frequent (*Figure 5A, C*). We also performed experiments with *L. japonicus* plants containing *pNIN:GUS* which exhibit β-glucuronidase (GUS) activity when the *NIN* (*NODULE INCEPTION*) promoter activity is induced by NF signaling (*Schauser et al., 1999*). GUS staining of roots showed that the *NIN* promoter activity was stronger in response to NGR234Δ*nopT* inoculation relative to inoculation by wild-type NGR234 (*Figure 5B, D*). Likewise, inoculation with NGR234Δ*nopT* resulted in an increased number of nodule primordia (*Figure 5G*). In contrast, over-expression of NopT under the control of the T7 promoter in the NGR234 parent strain reduced the number of nodule primordia when compared to inoculation with the wild-type strain (*Figure 5G*). These findings indicate that NopT negatively influences rhizobial infection and nodule initiation. Interestingly, the internal structures of nodules stained by toluidine blue dye exhibited remarkable similar when inoculated with both NGR234 and NGR234Δ*nopT* (*Figure 5—figure supplement 1*).

To confirm that the massive infection of roots by NGR234Δ*nopT* is due to the loss of NopT, we complemented the NGR234Δ*nopT* by expressing wild-type NopT. The *L. japonicus* plants inoculated with this strain had fewer infection foci, indicating restoration of the wild-type phenotype (*Figure 5E*). We also tested whether the proteolytic activity of NopT affected rhizobial infection. The protease-dead version of NopT$^{C93S}$ and three NopT forms with phospho-mimetic residues (i.e., NopT$^{T97D}$, NopT$^{S119D}$, and NopT$^{T257D}$ lacking proteolytic activity) were individually expressed in NGR234Δ*nopT*. Compared with the high number of infection foci induced by NGR234Δ*nopT* in *L. japonicus* plants, inoculation with these strains resulted in significantly fewer infection foci, but more than with the wild-type strain (*Figure 5E*).

We also expressed *nopT* under the control of a ubiquitin promoter in hairy roots of *L. japonicus* and inoculated the transgenic roots with DsRed-labeled *M. loti* MAFF303099, which does not possess a *nopT* gene in its genome. As shown in *Figure 5F*, reduced numbers of infection foci and infection threads were observed in the NopT expressing roots, indicating that NopT negatively affects the symbiosis between *L. japonicus* and *M. loti* strain.

To investigate whether the cleavage of NFR5 by NopT reduces rhizobial infection of *L. japonicus* roots, *NFR5* and *NFR5$^{5m}$* (uncleavable variant with five amino acid substitutions in the JM domain; *Figure 3F*) under the control of the native *NFR5* promoter, were expressed in hairy roots of *NFR5* knockout mutant plants (*nfr5-3*). NFR5 expression resulted in numerous infections and a significant increase of infection foci was observed upon inoculation with the NGR234Δ*nopT* mutant (*Figure 5H*). However, NFR5$^{5m}$ expression showed no effects on rhizobial infection (*Figure 5—figure supplement*

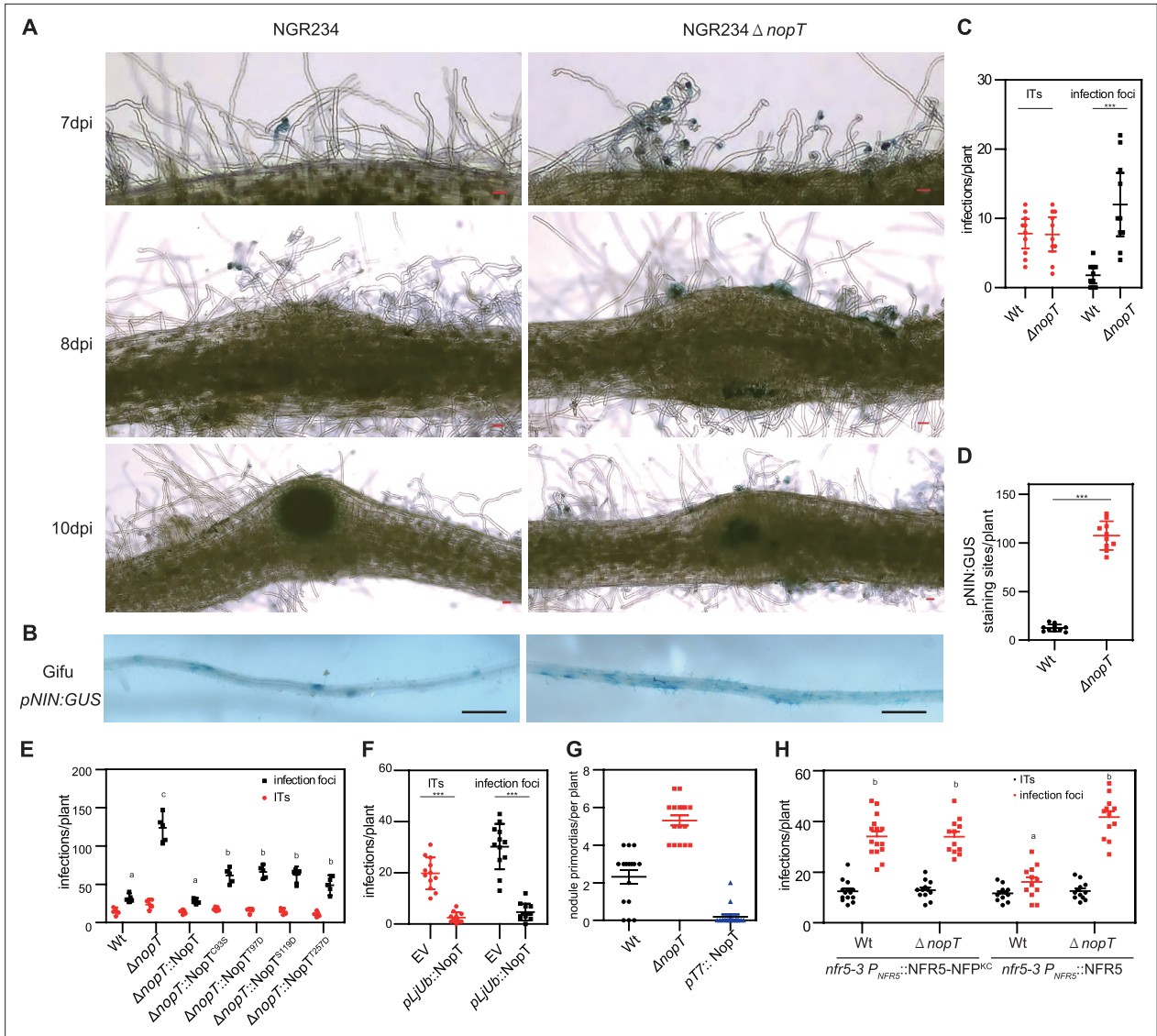

**Figure 5.** NopT regulates rhizobial infection in *L. japonicus*. (**A**) Analysis of rhizobial infection in *L. japonicus* roots inoculated with GUS-labeled *S. fredii* NGR234 (wild-type; WT) or a *nopT* knockout mutant (NGR234Δ*nopT*; abbreviated as Δ*nopT* in other panels). Scale bar = 100 μm. Infection represents both infection focus and infection thread. The infection threads in the root hairs were shown in the left image in the upper panel, while the infection foci were shown in the left image in the middle panel. (**B**) GUS staining pictures showing roots of *L. japonicus* plants expressing GUS expression under control of the *NIN* promoter (*pNIN:GUS*). Plants were inoculated with NGR234 or NGR234Δ*nopT* and analyzed at 7 dpi. Scale bar = 1 mm. (**C**) Infection data (ITs, infection threads) for roots shown in (**A**) at 7 dpi (*n* = 10, Student's *t*-test; *** indicates statistically significant differences at p < 0.01). (**D**) Quantification of GUS staining sites for roots shown in (**B**) (*n* = 10, Student's *t*-test, p < 0.01). (**E**) Infection data for wild-type roots inoculated with NGR234 (WT), NGR234Δ*nopT*, and NGR234Δ*nopT* expressing indicated NopT variants at 7 dpi (*n* = 5, Student's *t*-test: p < 0.01). (**F**) Analysis of rhizobial infection in hairy roots of *L. japonicus* (wild-type) expressing GFP (EV, empty vector control) or NopT. Plants were inoculated with DsRed-labeled *M. loti* MAFF303099 and analyzed at 5 dpi (*n* = 8, Student's *t*-test: p < 0.01). (**G**) Nodule primordia formation in *L. japonicus* wild-type roots inoculated with NGR234 (WT), NGR234Δ*nopT*, or NGR234 over-expressing *nopT* (*pT7*:NopT). Roots were analyzed at 14 dpi (*n* = 16, Student's *t*-test: p < 0.01). (**H**) Expression of NFR5 and NFR5-NFP^KC in hairy roots of *nfr5-3* mutant plants. Plants were inoculated with GFP-labeled NGR234 (WT) or NGR234Δ*nopT* and analyzed at 8 dpi. Roots expressing NFR5-NFP^KC showed high numbers of infection foci for both strains whereas significant differences were observed for NFR5 expressing roots (*n* > 10, Student's *t*-test: p < 0.01).

The online version of this article includes the following figure supplement(s) for figure 5:

**Figure supplement 1.** Cross-section images of nodules at 14 dpi after toluidine blue staining.

**Figure supplement 2.** Rhizobial infection in the mutant versions of NFR5.

2), suggesting that the amino acid substitutions in NFR5$^{5m}$ caused a conformational change that rendered the protein inactive in symbiotic signaling. As the NFR5$^{JM}$-NFP$^{KC}$ fusion protein was not cleaved by NopT (*Figure 3G*), we also constructed hairy roots of *nfr5-3* mutant plants in which *NFR5-NFP$^{KC}$* (NFR5 variant with its KC domain replaced by the KC of MtNFP) was expressed under the control of the *NFR5* promoter. Remarkably, inoculation of these plants showed no difference between NGR234Δ*nopT* and the NGR234 wild-type strain in terms of rhizobial infection foci (*Figure 5H*), suggesting that the cleavage of NFR5 by NopT reduced the degree of rhizobial infection. Taken together, these data indicate that the protease activity of NopT, its phosphorylation status and the proteolysis of the NFR5 target are important for the regulation of rhizobial infection.

## NopT from other *S. fredii* strains also cleave NFR5

Since NopT of *S. fredii* NGR234 (NopT$_{NGR234}$) cleaves NFR5, we wondered whether homologs from other rhizobial species also possess such a proteolytic activity. Phylogenetic analysis showed that NopT proteins of *Sinorhizobium* and *Bradyrhizobium* are located in different clades (*Figure 6—figure supplement 1*). *B. diazoefficiens* USDA110, a typical *Bradyrhizobium* strain used to study nodulation of soybeans, produces two NopT proteins (*Fotiadis et al., 2012*). In contrast to NopT$_{NGR234}$, however, neither NopT1$_{USDA110}$ nor NopT2$_{USDA110}$ were able to cleave co-expressed NFR5 in *E. coli* cells (*Figure 6A*). This finding suggests that NopT homologs from *Bradyrhizobium* species have lost the ability to cleave NFR5 and probably act on other, unknown host target proteins. The genome of various *Sinorhizobium* species (e.g., *S. fredii* USDA257 and *S. fredii* HH103) possess genes that encode truncated NopT forms, which are almost identical to the autocleaved version of NopT$_{NGR234}$ (*Khan et al., 2022*). Remarkably, NopT$_{USDA257}$ and NopT$_{HH103}$ were both able to cleave NFR5 *in E. coli* cells (*Figure 6A*). However, NopT$_{USDA257}$ were unable to suppress cell death triggered by NFR1 and NFR5 expression in *N. benthamiana* leaves (*Figure 6B*). Moreover, expression of NopT$_{USDA257}$ in NGR234Δ*nopT* had no effect on the infection ability of NGR234Δ*nopT*, as observed in inoculation tests with the *L. japonicus* line containing *pNIN:GUS* (*Figure 6C, D*). These results suggest that NopT$_{USDA257}$ and NopT$_{HH103}$ exhibit similar proteolytic activity to NopT$_{NGR234}$, but target different host proteins to regulate rhizobial infection.

## Discussion

Plant–microbe interactions are complex, employing distinct 'chemical weapons' or strategies to communicate mutually, thereby securing more resources for their survival (*Jones and Dangl, 2006*; *Schubert et al., 2020*). T3SS effectors of phytopathogenic bacteria are injected into plant cells to facilitate bacterial infection by targeting host proteins of the plant immune system, thereby suppressing PTI (*Jones and Dangl, 2006*; *Tang et al., 2017*). However, the role of rhizobial T3SS effectors in regulating the mutualistic symbiosis between legumes and rhizobia remains largely unknown. The aim of this study was to gain essential insights into the regulation mechanisms by which rhizobial effectors act directly on symbiotic signaling pathways to promote or dampen infection of host cells.

Understanding how plants recognize different microbes as 'friends' or 'foes' is a central question in biology. The perception of microbial signals by plant receptor proteins is essential for triggering specific signaling pathways to resist invading pathogens or to establish symbiosis with beneficial microbes (*Zipfel and Oldroyd, 2017*). Direct targeting of these receptors by T3SS effectors represents an efficient strategy of phytopathogenic bacteria to suppress PTI and promote bacterial infection. In the well-studied *Arabidopsis–P. syringae* interaction, T3SS effectors target plant flagellin receptor FLS2 and its coreceptor BAK1 (BRI1-[Brassinosteroid insensitive1]-associated receptor kinase 1) to suppress flagellin-induced PTI (*Zipfel et al., 2004*; *Sun et al., 2013*). For example, AvrPto targets FLS2 and HopB1 cleaves BAK1 (*Xiang et al., 2008*; *Li et al., 2016*). Likewise, AvrPtoB, an E3 ligase in *P. syringae*, targets FLS2 to promote its degradation and suppress PTI (*Göhre et al., 2008*). Despite the evolutionary importance of T3SS effectors in targeting host receptors, we were surprised to find that NopT, a rhizobial member of the C58 protease effector family, can directly target NFRs and manipulate the symbiotic signaling pathway in legumes. The interaction between NopT of *S. fredii* NGR234 and NFRs of *L. japonicus* was identified by using a unique screening system, in which co-expression of NFR1/NFR5 in *N. benthamiana* leaves leads to cell death. NopT, but not other effectors of NGR234, could suppress this cell death response. Consistent with these initial observations, inoculation of *L.*

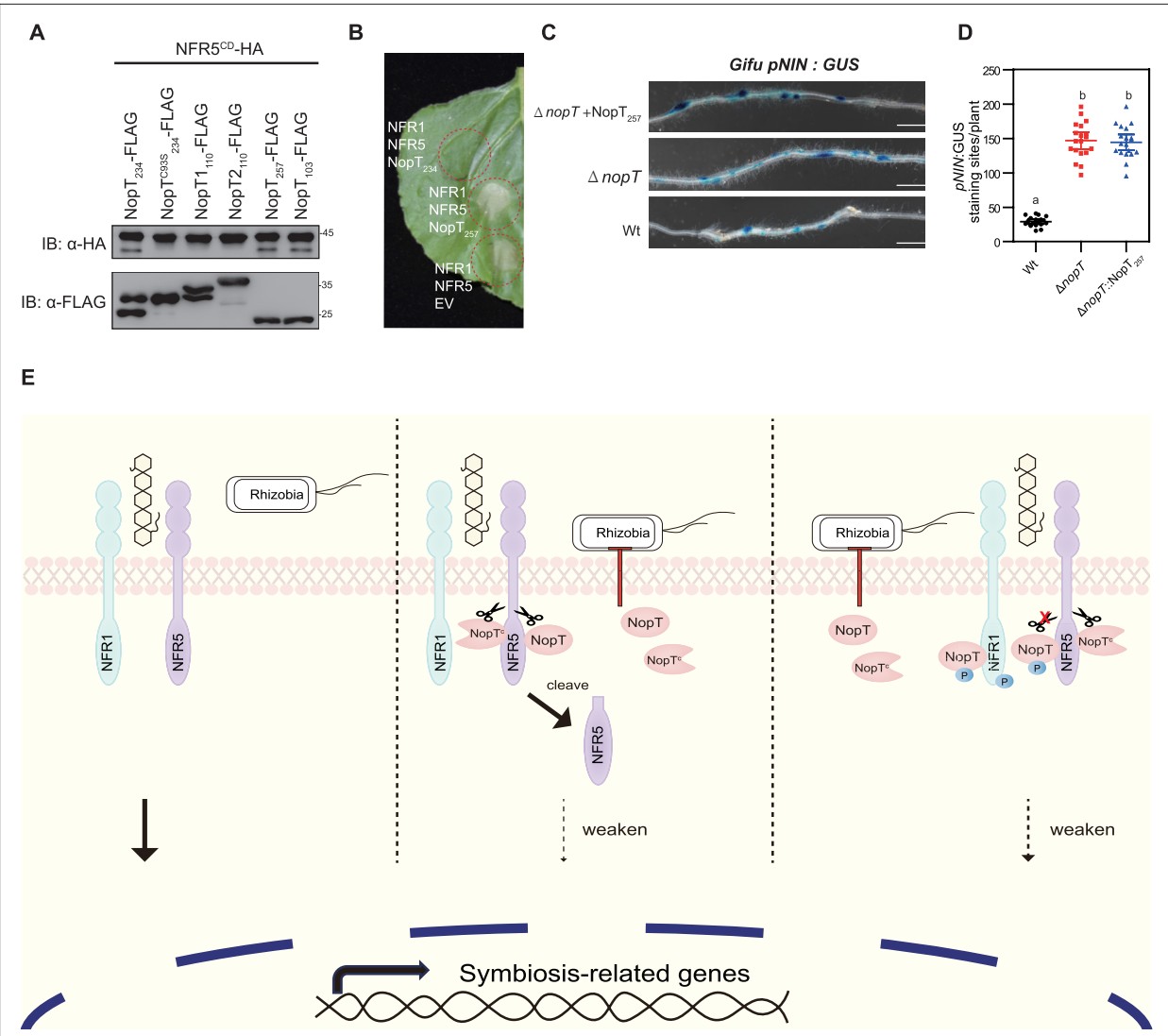

**Figure 6.** *S. fredii* NopT proteins cleave NFR5 and working model for NopT of NGR234. (**A**) NopT of *S. fredii* NGR234 and homologs from other rhizobial strains were co-expressed with NFR5$^{CD}$ in *E. coli* cells (NopT$_{234}$, NopT of NGR234; NopT1$_{110}$ and NopT2$_{110}$, NopT proteins of *B. diazoefficiens* USDA110; NopT$_{257}$, NopT of *S. fredii* USDA257; NopT$_{103}$, NopT of *S. fredii* HH103). Immunoblot analysis indicated NFR5 cleavage by NopT proteins from *S. fredii* strains. (**B**) Expression of NopT$_{257}$ in *N. benthamiana* could not inhibit the cell death triggered by co-expressed NFR1 and NFR5. (**C**) *L. japonicus* Gifu *pNIN:GUS* plants were inoculated with *S. fredii*, NGR234, NGR234Δ*nopT*, and NGR234Δ*nopT* expressing NopT$_{257}$ (Δ*nopT* + NopT$_{257}$). Roots were subjected to GUS staining at 7 dpi. Scale bar = 2.5 mm. (**D**) Quantitative analysis of GUS-stained roots shown in panel C (*n* = 19, Student's *t*-test: p < 0.01). (**E**) A proposed model for NopT of NGR234 interacting with NFRs. NopT and NopT$^C$ (autocleaved NopT) proteolytically cleave NFR5 at the juxtamembrane (JM) domain to release the intracellular domain of NFR5 (cleaved NFR5). NFR1 phosphorylates full-length NopT to block its proteinase activity. NopT$^C$ cannot be phosphorylated by NFR1.

The online version of this article includes the following source data and figure supplement(s) for figure 6:

**Source data 1.** Original files for western blot analysis displayed in *Figure 6A*.

**Source data 2.** PDF file containing original western blots for *Figure 6A*, indicating the relevant bands and treatments.

**Figure supplement 1.** Phylogenetic tree based on the amino acid sequence of NopT homologs from different bacterial species.

**Figure supplement 2.** The cleavage efficiency of NFR5 in different system.

**Figure supplement 3.** Transient expression of NopT triggers cell death in *Arabidopsis thaliana* and *Nicotiana tabacum*.

*japonicus* with the *S. fredii* NGR234Δ*nopT* mutant resulted in increased infection of *L. japonicus* roots. Moreover, over-expression of NopT in the roots was found to reduce rhizobial infection by *M. loti*. The interaction between NopT and NFR1/NFR5 was verified by BiFC, Split-LUC, and co-IP experiments.

Interestingly, using different assays, we found that NopT cleaves NFR5 at the JM domain. However, the efficiencies of NopT cleavage of NFR5 in *L. japonicus* and *N. benthamiana* were slightly different (*Figure 6—figure supplement 2*), which could be related to the uncontrollable activation of NFRs in *N. benthamiana* or phosphorylation of NopT by another kinase leading to decreased proteolytic activity. The modification of five conserved residues in the JM domain and the replacement of NFR5$^{KC}$ by NFP$^{KC}$ resulted in NFR5$^{CD}$ forms that were resistant to proteolysis by NopT in *E. coli* cells. These results suggest that cleavage of NFR5 depends on both the JM domain and the KC region. Although belonging to the same protease family, the NopT cleavage site in NFR5 was found to be polybasic as in the YopT substrate (*Shao et al., 2003b*; *Schmidt, 2011*), whereas the cleavage site of AvrPphB substrate is canonical with seven adjacent amino acids involved (*Kim et al., 2016*). These differences reflect a broad variety of YopT-type effector proteases and their substrates in host cells. Our study shows that NopT targets the NFR1/NFR5 complex thus reduces NF signaling and formation of nodule primordia in *L. japonicus*. NopT–NFR interactions may reflect a regulation mechanism to prevent rhizobial hyperinfection in host species other than *L. japonicus*.

Similar to the *Pseudomonas* effector AvrPphB, NopT of *S. fredii* NGR234 is able to cleave soybean PBS1-1 and expression of this effector in *S. fredii* USDA257 negatively influences symbiosis with certain soybean cultivars (*Khan et al., 2022*). Indeed, NopT of NGR234 acts as an Avr effector in some nonhost plants. Transient expression of NopT of strain NGR234 in *Arabidopsis* and tobacco triggers strong immune responses and cell death (*Dai et al., 2008*; *Kimbrel et al., 2013*; *Khan et al., 2022*; *Figure 6—figure supplement 3*). In *Arabidopsis*, NopT-induced cell death was found to be dependent on the PBS1-1 kinase and the resistance protein RPS5 (*Khan et al., 2022*). Likewise, when constitutively expressing NopT and NopT$^{C93S}$ in *L. japonicus*, we only obtained stable transgenic lines expressing NopT$^{C93S}$, indicating that the protease activity of NopT had a negative effect on plant development. In contrast, generation of hairy roots expressing NopT was possible. Such differences may be explained by different NopT substrates in roots and aerial parts of the plant. Indeed, our study shows that NopT not only cleaves NFR5 but is also able to proteolyze *Arabidopsis* AtLYK5 and *L. japonicus* LjLYS11. These two receptors possess chitin-binding affinities and trigger PTI responses (*Cao et al., 2014*; *Gysel et al., 2021*). NFR5, AtLYK5, and LjLYS11 are kinase-dead LYKs belonging to the same LYK subfamily. Thus, NopT appears not only to suppress NF signaling, but may also interfere with signal transduction pathways related to plant immunity. Depending on the host legume species, NopT could suppress PTI or induce ETI, thereby modulating rhizobial infection and nodule formation. Interactions between NopT and proteins related to the plant immune system may represent an important evolutionary driving force for host-specific nodulation and explain why the presence of NopT in NGR234 has a negative effect on symbiosis with *L. japonicus* but a positive one with other legumes.

The regulation of kinase-dead pseudokinases in plants may be more complex compared to the well-studied control of protein kinase activities by phosphorylation. The cleavage of receptor-like protein kinase may be an overlooked strategy to regulate their function in plant biology. Regulation of protein activities by cleavage is well studied in animals, for example, for Notch signaling, where proteolysis of Notch molecules leads to downstream signaling (*Kopan and Ilagan, 2009*). *Arabidopsis* BAK1 undergoes proteolytic cleavage, which is important for both brassinosteroid signaling and induction of PTI responses (*Zhou et al., 2019*). Whereas, HopB1 cleaves immune co-receptor BAK1 at the kinase domain to inhibit plant defense (*Li et al., 2016*). NFR5 is a pseudokinase that is essentially required for the activation of NF signaling. However, the mechanisms of how the pseudokinase domain of NFR5 mediates symbiotic signaling transduction remains unclear. Our work indicates that NopT proteolytically cleaves NFR5 and suppresses NF signaling. Such a fine-tuning process could be advantageous for the symbiosis in host plants other than *L. japonicus*, for example to prevent hyperinfection. However, the precise function of NFR5 cleavage by NopT and the fate of the released cytoplasmic domain remain to be clarified.

Mutual regulation between T3SS effectors and host target proteins has been investigated for various plant–pathogen interactions. For example, the conserved *Pseudomonas* effector AvrPtoB can directly ubiquitinate the key components of plant immunity to promote bacterial virulence (*Göhre*

*et al., 2008*). However, the activity of AvrPtoB in promoting virulence could either be enhanced or dampened by host proteins. AvrPtoB can be phosphorylated by SnRK2.8 to promote its virulence activity (*Lei et al., 2020*). On the other hand, the plant lectin receptor-like kinase LexRK-IX.2 and the Pto kinase phosphorylate AvrPtoB to inactivate its ubiquitin E3 ligase activity and undermine the effector's ability to suppress PTI (*Ntoukakis et al., 2009*; *Xu et al., 2020*). A similar strategy is used by NFR1 to inhibit the protease activity of NopT via phosphorylation. NFR1 expressed in *E. coli* cells phosphorylates full-length NopT of *S. fredii* NGR234 at several residues. Three NopT forms with a phosphomimic substitution (i.e., NopT$^{T97D}$, NopT$^{S119D}$, and NopT$^{T257D}$) lacked autoproteolytic activity and the ability to cleave NFR5. The expression of these NopT variants in the NGR234Δ*nopT* mutant showed little effects in comparison with expression of wild-type NopT. These data suggest that *L. japonicus* utilizes NFR1 to phosphorylate NopT in order to dampen its catalytic activity and protect NFR5 from cleavage, which favors rhizobial infections. However, NopT$^{ΔN50}$, which is similar to autocleaved NopT, retained the ability to interact with NFR5 but not with NFR1 (*Figure 2—figure supplement 2*). Moreover, full-length NopT, but not autocleaved NopT, migrated slower on gels when NFR1$^{CD}$ was co-expressed in *E. coli* cells, suggesting that autocleaved NopT may escape being phosphorylated and inactivated by NFR1. Future studies are required to explore the interactions between acylated NopT and NFR1. When expressed in *N. benthamiana*, GFP-tagged NopT forms lacking their autocleavage site (and thus non-acylated) were found to be localized within the cytoplasm, whereas NopT was targeted to the plasma membrane (*Khan et al., 2022*). Accordingly, NopT without acylation sites showed weaker interactions with NFRs in our study, suggesting that acylation of NopT, leading to plasma membrane location, promotes NopT–NFR interactions.

NopT homologs vary in different *S. fredii* strains. USDA257 and HH103 produce truncated NopT versions different from NGR234 due to a 19-bp deletion (*Khan et al., 2022*). Accordingly, NopT of these strains lacks autocleavage and acylation sites and possess a different N-terminal secretion signal sequence (*Akeda and Galán, 2005*). Nevertheless, NopT$_{HH103}$ is a functional T3SS effector as mutant analysis indicated a symbiosis-promoting role in certain soybean cultivars (*Li et al., 2023*). Similar to NopT$_{NGR234}$, NopT$_{USDA257}$, and NopT$_{HH103}$ were able to cleave NFR5 in *E. coli* cells. Since NopT$_{USDA257}$ and NopT$_{HH103}$ cannot be acylated, they likely accumulate in the cytoplasm of host legumes and are therefore less efficient in cleaving plasma membrane-bound NFR5 in comparison to acylated NopT$_{NGR234}$. This could also explain why NopT$_{USDA257}$ was unable to suppress cell death triggered by NFR1/NFR5 in *N. benthamiana* leaves. It can be expected that NFR1 does not interact with and phosphorylate NopT$_{USDA257}$ or NopT$_{HH103}$, as these effectors are similar to NopT$^{ΔN50}$ of NGR234, which cannot interact with NFR1. Consistent with these findings, expression of NopT$_{USDA257}$ in NGR234Δ*nopT* did not alter the infection phenotype of NGR234Δ*nopT*. It can be hypothesized that the host targets of these truncated NopT effectors might be different from those of NopT$_{NGR234}$. *Bradyrhizobium* NopT effectors are also different from NopT$_{NGR234}$, as NopT of *B. diazoefficiens* USDA110 was unable to cleave NFR5. NopT effectors may play a crucial role in *Bradyrhizobium*–legume interactions. A *Bradyrhizobium* sp. ORS3257 mutant deficient in NopT production induced only ineffective nodules on roots of *A. indica* and most of the formed nodules were not infected (*Teulet et al., 2019*). Overall, these findings indicate versatile functions of NopT homologs in different strains and suggest that NopT controls different regulation processes in specific host cells.

We present here a model of mutual regulation, in which NopT proteolyzes NFR5 to suppress NF signaling, whereas protease activity of full-length NopT is suppressed by NFR1 via phosphorylation (*Figure 6E*). NopT impairs the function of the NFR1/NFR5 receptor complex. Cleavage of NFR5 by NopT reduces its protein levels. Possible inhibitory effects of NFR5 cleavage products on NF signaling are unknown but cannot be excluded. Inactivation of NopT protease activity via phosphorylation by kinases such as NFR1 appears to be a countermeasure of the host to enable symbiotic signaling and rhizobial infection. Such feedback regulation could be the result of evolutionary pressure to produce 'improved' NFRs, which are resistant to cleavage by NopT and have an increased capacity to phosphorylate NopT. On the other hand, suppression of NopT protease activity by NFRs may drive broad-host-range strains such as NGR234 to produce higher levels and structurally modified NFs.

The negative regulation of plant symbiotic signaling by NopT is detrimental to rhizobial infection, which appears paradoxical from an evolutionary perspective. Two possible scenarios will be discussed. One possibility is that NopT may have originated from its homologs, such as AvrPphB, and possesses a pathogenic feature by proteolyzing immune receptors to disrupt the plant immune pathway.

However, due to sequence similarity, NFR5 is also proteolyzed by NopT. Expression of NopT leading to strong cell death in *N. tabacum* and *Arabidopsis* also suggests that NopT might be recognized as Avr effector to trigger strong immunity. Another possibility is that the presence of NopT serves as a strategy for rhizobia to evade detection by plant immunity, which may be activated by rhizobial over-infection. The negative effects on symbiosis by cleaving NFR5 is then inhibited by NFR1 via a direct phosphorylation. In a parallel study, *S. fredii* NGR234 NopM was also identified to interact with and mediate the ubiquitination of NFR5, stabilizing NFR5 levels and thereby promoting rhizobial infection and nodulation (*Wang et al., 2024*). Therefore, both *S. fredii* NGR234 NopT and NopM target the same protein and appear to antagonistically regulate rhizobial symbiosis; however, the exact role of NopT in nodulation needs to be elucidated in future research.

Taken together, this study provides insights into a legume–rhizobium interaction, in which the bacterium deploys an effector protease to dampen symbiotic signaling, while the host plant counter-acts by phosphorylating the effector, leading to its inactivation. Our findings highlight the function of a bacterial effector protease in regulating a symbiotic signaling pathway in legumes. This opens up the perspective of developing specific kinase-interacting proteases to reprogram and fine-tune cellular signaling processes in general.

# Materials and methods

## Germination and growth of *L. japonicus*

*L. japonicus* (ecotype Gifu B-129) was provided by the Center for Carbohydrate Recognition and Signaling (https://lotus.au.dk/), and used as wild-type for nodulation assays. The Gifu *pNIN:GUS* transgenic line and *nfr* mutants (*nfr1-1* and *nfr5-3*) were kindly provided by Dr. Jens Stougaard from the Aarhus University, Denmark (*Madsen et al., 2003*; *Radutoiu et al., 2003*; *Heckmann et al., 2011*). All *L. japonicus* seeds were treated with concentrated sulfuric acid for 10 min, followed by surface sterilization in 1% (wt/vol) NaClO for 8 min. After incubation at 4°C in the dark for 2 days, the seeds were placed on 0.8% (wt/vol) agar containing half-strength Murashige & Skoog (MS) medium for germination at 22°C for 2 days in the dark. Seedlings were then transferred to growth pots containing vermiculite supplied with half-strength B&D (Broughton & Dilworth) medium without nitrogen under long-day conditions (16 hr light/8 hr dark) at 22°C. Ten-day-old seedlings were inoculated with rhizobial cultures ($OD_{600}$ = 0.02) for infection experiments.

## Plasmid construction

All plasmids used in this study were generated using MultiF Seamless Assembly Mix (Abclonal, # RK21020). The plasmid pGWB514 was used as an original plasmid for construction of all binary vectors (*Nakagawa et al., 2007*). Briefly, different tags including HA tag, Myc tag, FLAG tag, Strep tag, NLuc, CLuc, and GFP were amplified using a forward primer containing a *Kpn*I site and ligated into pGWB514 digested with *Xba*I and *Sac*I using the seamless cloning method described above. The generated plasmids were named pG5XX-HA, pG5XX-Myc, pG5XX-FLAG, pG5XX-NLuc, pG5XX-CLuc, and pG5XX-GFP, respectively. The coding sequences of *L. japonicus* *NFR1* and *NFR5* were cloned into pGWB514 and pG517 between *Xba*I/*Kpn*I to generate NFR1-HA and NFR5-Myc fusion constructs under the control of the cauliflower mosaic virus 35S promoter. The DNA fragment containing *pro35S*:NFR5-Myc-NosT was then amplified and cloned into pG514-NFR1 at *Sbf*I. The final plasmids contained *pro35S*:NFR1-HA-NosT and *pro35S*:NFR5-Myc-NosT. Rhizobial genes encoding effectors from *S. fredii* NGR234 were cloned into pG5XX-FLAG between *Xba*I/*Kpn*I. To detect the interaction between NFR1/NFR5 and NopT in *N. benthamiana*, the full-length *NFR1/NFR5*, *NopT*, and *AtFLS2* coding sequences were cloned into pG5XX-NLuc, pG5XX-CLuc, pSPYCE(MR), and pSPYNE(R)173 (*Waadt et al., 2008*). For cleavage assays in *N. benthamiana* leaves, the *NFR5* and *NopT* coding sequences were cloned into pG5XX-FLAG, pG5XX-GFP, and pG5XX-Strep, respectively. For cleavage assays in *E. coli* cells, the sequence encoding the cytoplasmic domain of NFR5 was cloned into pACYC duet Strep-HA (*Han et al., 2017*), and the sequences encoding the kinase domain of NFR5 fused to the HA tag and the JM domain of NFR5 fused to the GFP tag were cloned into pSUMO. The point mutations in *NFR5^CD* and *NopT* were created by site-directed mutagenesis PCR using the templates pACYC duet Strep-NFR5^CD-HA and pET28a NopT-FLAG. For NFR/NopT cleavage experiments with *L. japonicus* plants, NopT-FLAG or NopT^C93S-FLAG was cloned into pUB-GFP between *Xba*I and *Kpn*I,

and 35S-NFR5-Myc-NosT was cloned into pUB-NopT-GFP at *Pst*I. For **Figure 3—figure supplement 1** NopT phosphorylation experiments with *E. coli*, the sequence encoding the cytoplasmic domain of NFR1 fused to a Myc tag was cloned into pCDF duet. For complementation experiments with the NGR234Δ*nopT* mutant (**Dai et al., 2008**), a 1317-bp DNA fragment upstream of *NopT* and the coding sequence of a given *NopT* sequence was fused using overlap PCR and cloned into pHC60 between *Xho*I and *Kpn*I. For complementation experiments with the *nfr5-3* mutant, a 1316-bp *NFR5* promoter DNA fragment was fused to NFR5-NFP$^{KC}$ or NFR5$^{5m}$ using overlap PCR and cloned into pUB-Cherry between *Pst*I and *Kpn*I. Primers used in this study are indicated in **Supplementary file 1B**.

## Transient gene expression in *Nicotiana* leaves

The plasmids of *INF1*, *Avr3*, *BAX*, and *R3a* for transient expression in *N. benthamiana* were kindly gifted by Dr. Juan Du from Huazhong Agricultural University. *Agrobacterium tumefaciens* strain EHA105 carrying various constructs were mixed with an *Agrobacterium* culture harboring the P19 suppressor in an infiltration buffer (10 mM MgCl$_2$, 10 mM MES-KOH pH = 5.8 and 200 µM acetosyringone). After incubation for 2 hr at room temperature, *Agrobacterium* cultures were infiltrated into the leaves of 4-week-old *N. benthamiana* and *N. tabacum* plants. Leaves were harvested and analyzed 2-day post inoculation. The cell death suppression experiments with *N. benthamiana* plants were performed as described previously (**Wang et al., 2011**).

## Hairy root transformation

Hairy root transformation of *L. japonicus* plants was performed as described previously (**Li et al., 2018**). In brief, surface-sterilized seeds were germinated and grown on MS plates without sucrose (23°C in the dark for the first 3 days and 23°C at a 16-hr light/8-hr dark period for the following 2 days).The seeds were then cut at the middle of the hypocotyl and co-cultivated with *A. rhizogenes* LBA1334 carrying pUB-GFP, pUB-NopT-FLAG-GFP, pNFR5:NFR5-mCherry, pNFR5:NFR5$^{5m}$-mCherry, and pNFR5:NFR5-NFP$^{KC}$-mCherry, respectively (23°C in the dark for the first 3 days and 23°C at a 16-hr light/8-hr dark period for the following 2 days). The plants were then transferred onto solid B5 medium. After about 15 days, non-fluorescent hairy roots (lacking expression of GFP or mCherry) were removed. The plants were inoculated with rhizobia: *S. fredii* NGR234, NGR234Δ*nopT*, or *Mesorhizobium loti* MAFF303099, and analyzed at indicated time points.

## Analysis of NFR5 cleavage by NopT and variants

In experiments with *E. coli*, Strep-NFR5$^{CD}$-HA or Strep-NFR5$^{CD}$-HA variants were co-expressed with NopT-FLAG- or FLAG-tagged NopT variants or AvrPphB-FLAG by using Duet vectors (Novagen; **Han et al., 2017**). Bacterial cultures were treated with 0.5 mM IPTG at 22°C for 15 hr. After washing the bacterial cells with PBS buffer, the bacterial pellet was resuspended in extraction buffer consisting of Tris-HCl (pH = 7.3), 150 mM NaCl, 5 mM EDTA, 0.5% (wt/vol) SDS and 0.5% (vol/vol) Triton X-100. After adding SDS loading buffer and boiling at 100°C for 5 min, the extracted proteins were subjected to immunoblot analysis using anti-HA-peroxidase (Sigma, clone 3F10) or anti-FLAG (Sigma, F1804) antibodies. An anti-groEL antibody (ABclonal,, A0969) was used to confirm equal loading of proteins on gels.

For in vitro cleavage assays, His-SUMO-NFR5$^{JM}$-GFP and NopT-FLAG-His proteins were expressed in *E. coli* cells and purified. The reaction mixture contained 3 µg NopT-FLAG (or NopT$^{C93S}$-FLAG) and 1 µg SUMO-NFR5$^{JM}$-GFP in extraction buffer without SDS and EDTA. After incubation at 22°C for 7 hr, 10 µl of a Ni-charged resin (GenScript, Cat. No. L00223) were added, and the cleavage products in the flow through were detected by immunoblot analysis using an anti-GFP antibody (ABclonal, AB_2770402).

To analyze cleavage of NFR5 in *L. japonicus* roots, hairy roots co-expressing NopT-FLAG or NopT$^{C93S}$-FLAG with NFR5-Myc were harvested at about 15 days after induction of hairy roots without rhizobia inoculation and subjected to immunoblot analysis. Expressed proteins were extract by extraction buffer and immunodetected with anti-FLAG antibody and anti-Myc (Bio-Legend, # 626808) antibody.

For NFR5/NopT cleavage experiments with *N. benthamiana* plants, leaves expressing proteins (NopT-FLAG or NopT$^{C93S}$-FLAG with NFR5-GFP) were analyzed. Leaf discs were collected at 2 days post infiltration with *agrobacteria* and extracted in the extraction buffer (Tris-HCl pH = 7.3, 150 mM NaCl, 5 mM EDTA, 0.5% (wt/vol) SDS and 0.5% (vol/vol) Triton X-100) and 2% protease inhibitors

(Sigma, P9599). NFR5 was immunodetected with anti-Myc and anti-GFP (ABclonal, AB_2770402) antibodies. The NopT or NopT[C93S] were detected with anti-FLAG antibody. The actin proteins were detected with anti-Actin antibody (Abclonal, # AC009).

## Phosphorylation assays

For NopT phosphorylation in *E. coli* cells, His-NFR1[CD]-Myc or His-NFR1[CD-K351E]-Myc was co-expressed with NopT-FLAG-His by using Duet vectors (Novagen). Protein expression was induced by 0.5 mM IPTG and cells were grown at 28°C for 18 hr. After simultaneous purification of NFR1[CD] and NopT using Ni-charged resin, the proteins were incubated with CIAP (TaKaRa, 2250B) at 37°C for 3 hr. The mobility shifts of NFR1[CD] and NopT were detected with anti-Myc (Bio-Legend, 626808) and anti-FLAG (Sigma, F1804) antibodies by immunoblot analysis.

In phosphorylation experiments with *L. japonicus* plants, NopT[C93S]-FLAG was expressed in hairy roots of the *nfr1-1* mutant and wild-type plants,, respectively. After removal of roots without fluorescence signals, plants were kept on half-strength B&D medium for 5 days. Roots were inoculated with *Mesorhizobium loti* MAFF303099 or treated with water. Total proteins were extracted using the extraction buffer (Tris-HCl pH = 7.3, 150 mM NaCl, 5 mM EDTA, 0.5% (wt/vol) SDS, and 0.5% (vol/vol) Triton X-100) supplemented with 2% protease inhibitor cocktail (Sigma, P9599) and 2% phosphatase inhibitor cocktail (Yeasen, 20109-A). Proteins were then precipitated with TCA to remove various contaminants (e.g., EDTA and surfactants). Phosphorylation of the NopT[C93S] protein was analyzed by 50 µM $Zn^{2+}$-Phos-tag SDS–PAGE gel (*Kato and Sakamoto, 2019*) and detected with the anti-FLAG antibody by immunoblot analysis. As loading control, actin proteins were immunodetected with an anti-Actin antibody.

## BiFC and Split-LUC assays

In the BiFC assay with *N. benthamiana* plants, the eYFP fluorescence of expressed fusion proteins was recorded 2–3 days post infiltration with *A. tumefaciens* using a confocal microscope (Leica TCS SP8). For the Split-LUC assay, *N. benthamiana* leaves were collected at 2 days post infiltration and sprayed with a solution of 1 mM D-luciferin (Promega, E1603) and 0.02% (vol/vol) Triton X-100. After a dark adaptation of 5 min, bioluminescence images were acquired using a Tanon Bio-Imaging System (Tanon, Shanghai, 4600).

## LC–MS analysis

For analysis of NopT phosphorylation sites, NopT-FLAG-His was co-expressed with His-labeled NFR1[CD] or NFR1[CD-K351E] in *E. coli* cells. The proteins were purified using Ni-charged resin (GenScript, Cat. No. L00223). The bands containing NopT[P] and NopT (control) proteins were cut out, destained by 50 mM triethylammonium bicarbonate (TEAB) solution (50% (vol/vol) acetonitrile in 50 mM TEAB), and washed with 100% acetonitrile until the gel turned white. The protein bands were then in-gel digested with sequencing-grade trypsin. LC–MS analysis was performed by Novogene Co, Ltd (Beijing, China).

For identification of the cleavage site in NFR5, NopT, and His-SUMO-NFR5[CD]-GST were co-expressed in *E. coli* cells and purified using Glutathione Resin (GenScript, # L00206). About 80 µg of the cleaved protein was collected for analysis. The protein bands were digested with trypsin, glu-C, chymotrypsin, and pepsin. LC–MS analysis was performed by Oulu Biotechnology Co, Ltd (Shanghai, China).

## Examination of rhizobial infection

The morphology of rhizobial infection foci and infection threads have been described previously (*Rae et al., 2021*). An infection focus is formed when the tip of a root hair curls around a single bacterium, forming an infection pocket. The infection thread is a tubular structure forming in the root hair, then extends to cortical cells. The number of infection foci and infection threads was determined using a Leica fluorescence microscope (DM2500). Whole roots were mounted on microscope slides and infection foci and infection thread counts were expressed on a per-root basis.

Histochemical GUS staining of *L. japonicus* Gifu B-129 or Gifu *pNIN:GUS* roots inoculated with different rhizobial strains was performed using the chromogenic substrate 5-bromo-4-chloro-3-indolyl-β-D-glucuronide as described (*Heckmann et al., 2011*). After evacuation of air for 20 min, root samples were incubated in GUS staining solution overnight (at 37°C in the dark). The GUS-stained

tissues were observed and photographed using a Motic Swift M200D compound microscope (Gifu B-129 roots inoculated with rhizobial strains harboring pT7-GUS) or a Nikon SMZ18 stereo microscope (Gifu *pNIN:GUS* inoculated with unlabeled rhizobia).

## Acknowledgements

We greatly thank Prof. Zhongming Zhang for his valuable suggestions on this project. We thank Drs. Eric Giraud, Ertao Wang, and Yan Liang for critical reading of this manuscript. We thank Dr. Jens Stougaard for kindly providing *L japonicus pNIN:GUS* and *nfr* mutant seeds. Dr. Juan Du is thanked for providing plasmids for expression of Avr3a and R3a, BAX, and INF1. The qPCR and microscopy data were acquired from the Core Facility Center at the National Key Lab of Agricultural Microbiology. This work was supported by the National Key R&D Program of China (2019YFA0904700), the National Natural Science Foundation of China (32090063), and the Fundamental Research Funds for the Central Universities (2662022SKYJ002 and 2662021JC010). Work in the Stacey's lab was supported by the NSF Plant Genome Research Program (2048410).

## Additional information

### Funding

| Funder | Grant reference number | Author |
| --- | --- | --- |
| National Key R&D Program of China | 2019YFA0904700 | Yangrong Cao |
| National Natural Science Foundation of China | 32090063 | Yangrong Cao |
| Fundamental Research Funds for the Central Universities | 2662022SKYJ002 | Yangrong Cao |
| Fundamental Research Funds for the Central Universities | 2662021JC010 | Yangrong Cao |
| National Science Foundation | Plant Genome Research Program 2048410 | Gary Stacey |

The funders had no role in study design, data collection, and interpretation, or the decision to submit the work for publication.

### Author contributions

Hanbin Bao, Conceptualization, Data curation, Formal analysis, Investigation, Methodology, Writing – original draft, Writing – review and editing; Yanan Wang, Data curation, Formal analysis, Investigation, Visualization, Methodology, Writing – original draft; Haoxing Li, Data curation, Investigation; Qiang Wang, Ying Ye, Investigation; Yutao Lei, Investigation, Methodology; Syed F Wadood, Methodology; Hui Zhu, Resources, Supervision; Christian Staehelin, Data curation, Writing – review and editing; Gary Stacey, Conceptualization, Funding acquisition, Writing – review and editing; Shutong Xu, Supervision, Methodology; Yangrong Cao, Conceptualization, Resources, Data curation, Formal analysis, Supervision, Funding acquisition, Writing – original draft, Project administration, Writing – review and editing

### Author ORCIDs

Hanbin Bao https://orcid.org/0000-0002-2950-1234
Yanan Wang https://orcid.org/0000-0001-5797-1113
Hui Zhu https://orcid.org/0000-0002-7273-8448
Gary Stacey https://orcid.org/0000-0001-5914-2247
Yangrong Cao https://orcid.org/0000-0002-2252-6551

Reviewer #1 (Public review): https://doi.org/10.7554/eLife.97196.4.sa1

Reviewer #2 (Public review): https://doi.org/10.7554/eLife.97196.4.sa2

Author response https://doi.org/10.7554/eLife.97196.4.sa3

## Additional files

### Supplementary files

Supplementary file 1. Phosphopeptide identification and oligonucleotide used in the study. (A) Phosphopeptides identified by liquid chromatography–mass spectrometry. (B) Oligonucleotides used in the study.

MDAR checklist

### Data availability

All data generated or analyzed during this study are included in the manuscript and supporting files.

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
