## [Editor Report · eLife Assessment]

This manuscript presents **important** findings on a bacterial effector involved in plant symbiotic signaling. The effector proteolytically targets a key receptor while its activity is counteracted by host-mediated phosphorylation, revealing a dynamic interplay that fine-tunes symbiotic interactions. The evidence supporting these claims is **solid**, and the findings have potential signaling implications beyond bacterial interactions with plants.

---

## [Referee Report · Reviewer #1 (Public review)]

Bacterial effectors that interfere with the inner molecular workings of eukaryotic host cells are of great biological significance across disciplines. On the one hand they help us to understand the molecular strategies that bacteria use to manipulate host cells. On the other hand, they can be used as research tools to reveal molecular details of the intricate workings of the host machinery that is relevant for the interaction/defence/symbiosis with bacteria. The authors investigate the function and biological impact of a rhizobial effector that interacts with and modifies, and curiously is modified by, legume receptors essential for symbiosis. The molecular analysis revealed a bacterial effectorthat cleaves a plant symbiosis signaling receptor to inhibit signaling and the host counterplay by phosphorylation via a receptor kinase. These findings have potential implications beyond bacterial interactions with plants. Bao and colleagues investigated how rhizobial effector proteins can regulate the legume root nodule symbiosis.

Bao and colleagues investigated how rhizobial effector proteins can regulate the legume root nodule symbiosis. A rhizobial effector is described to directly modify symbiosis-related signaling proteins, altering the outcome of the symbiosis. Overall, the paper presents findings that will have a wide appeal beyond its primary field.

Out of 15 identified effectors from Sinorhizobium fredii, they focus on the effector NopT, which exhibits proteolytic activity and may therefore cleave specific target proteins of the host plant. They focus on two Nod factor receptors of the legume Lotus japonicus, NFR1 and NFR5, both of which were previously found to be essential for the perception of rhizobial nod factor, and the induction of symbiotic responses such as bacterial infection thread formation in root hairs and root nodule development (Madsen et al., 2003, Nature; Tirichine et al., 2003; Nature). The authors present evidence for an interaction of NopT with NFR1 and NFR5. The paper aims to characterize the biochemical and functional consequences of these interactions and the phenotype that arises when the effector is mutated.

Evidence is presented that in vitro NopT can cleave NFR5 at its juxtamembrane region. NFR5 appears also to be cleaved in vivo, and NFR1 appears to inhibit the proteolytic activity of NopT by phosphorylating NopT. When NFR5 and NFR1 are ectopically over-expressed in leaves of the non-legume Nicotiana benthamiana, they induce cell death (Madsen et al., 2011, Plant Journal). Bao et al. found that this cell death response is inhibited by the coexpression of nopT. Mutation of nopT alters the outcome of rhizobial infection in L. japonicus. These conclusions are well supported by the data.

The presented data support the interaction of NopT with NFR1 and NFR5. In particular, there is solid support for cleavage of NFR5 by NopT (Figure 3) and the identification of NopT phosphorylation sites that inhibit its proteolytic activity (Figure 4C). Cleavage of NFR5 upon expression in N. benthamiana (Figure 3A) requires appropriate controls (inactive mutant versions), since Agrobacterium as a closely rhizobia related bacterium might increase defense related proteolytic activity in the plant host cells, and these controls are provided.

Key results from N. benthamiana appear consistent with data from recombinant protein expression in bacteria. For the analysis in the host legume L. japonicus transgenic hairy roots were included. To demonstrate that the cleavage of NFR5 occurs during the interaction in plant cells, the authors build largely on Western blots. Regardless of whether Nicotiana leaf cells or Lotus root cells are used as the test platform, the Western blots indicate that only a small proportion of NFR5 is cleaved when co-expressed with nopT, and most of the NFR5 persists in its full-length form (Figures 3A-D). The authors discuss how the loss of NFR5 function (loss of cell death, impact on symbiosis) can be explained despite this vast excess of intact NFR5, but do not further explore the impact of this ratio on downstream signaling.

---

## [Referee Report · Reviewer #2 (Public review)]

Summary:

This manuscript presents data demonstrating NopT's interaction with Nod Factor Receptors NFR1 and NFR5 and its impact on cell death inhibition and rhizobial infection. The identification of a truncated NopT variant in certain Sinorhizobium species adds an interesting dimension to the study. These data try to bridge the gaps between classical Nod-factor-dependent nodulation and T3SS NopT effector-dependent nodulation in legume-rhizobium symbiosis. Overall, the research provides interesting insights into the molecular mechanisms underlying symbiotic interactions between rhizobia and legumes.

Strengths:

The manuscript nicely demonstrates NopT's proteolytic cleavage of NFR5, regulated by NFR1 phosphorylation, promoting rhizobial infection in L. japonicus. Intriguingly, authors also identify a truncated NopT variant in certain Sinorhizobium species, maintaining NFR5 cleavage but lacking NFR1 interaction. These findings bridge the T3SS effector with the classical Nod-factor-dependent nodulation pathway, offering novel insights into symbiotic interactions.

Weaknesses:

(1) In the previous study, when transiently expressed NopT alone in Nicotiana tobacco plants, proteolytically active NopT elicited a rapid hypersensitive reaction. However, this phenotype was not observed when expressing the same NopT in Nicotiana benthamiana (Figure 1A). Conversely, cell death and a hypersensitive reaction were observed in Figure S8. This raises questions about the suitability of the exogenous expression system for studying NopT proteolysis specificity.

(2) NFR5 Loss-of-function mutants do not produce nodules in the presence of rhizobia in lotus roots, and overexpression of NFR1 and NFR5 produces spontaneous nodules. In this regard, if the direct proteolysis target of NopT is NFR5, one could expect the NGR234's infection will not be very successful because of the Native NopT's specific proteolysis function of NFR5 and NFR1. Conversely, in Figure 5, authors observed the different results.

(3) In Figure 6E, the model illustrates how NopT digests NFR5 to regulate rhizobia infection. However, it raises the question of whether it is reasonable for NGR234 to produce an effector that restricts its own colonization in host plants.

(4) The failure to generate stable transgenic plants expressing NopT in Lotus japonicus is surprising, considering the manuscript's claim that NopT specifically proteolyzes NFR5, a major player in the response to nodule symbiosis, without being essential for plant development.

Comments on the revised version:

My concerns regarding the potential function of NopT during nodule symbiosis have been adequately addressed in the revised manuscript. Therefore, I have no further questions about this version, aside from a few minor suggestions:

(1) Please carefully check the text formatting throughout the manuscript to ensure consistency with scientific conventions and the journal's standards. For example, Line 105-117 and line119-131.

(2) The term "detrimental" in line 624 may not accurately describe the function of NopT in rhizobial infection. Since the authors propose that NopT proteolytically cleaves NFR5 and suppresses NF signaling as a potential fine-tuning mechanism for legume symbiosis, a more precise term may be needed.

(3) Lines 632-634 are somewhat unclear. If NopT serves as a strategy for rhizobia to evade detection by plant immunity, then knocking out NopT should, in theory, inhibit rhizobial infection. Clarification on this point would be beneficial.

---

## [Author Response]

The following is the authors’ response to the previous reviews.

**Public Reviews:**

**Reviewer #1 (Public review):**
Bacterial effectors that interfere with the inner molecular workings of eukaryotic host cells are of great biological significance across disciplines. On the one hand they help us to understand the molecular strategies that bacteria use to manipulate host cells. On the other hand they can be used as research tools to reveal molecular details of the intricate workings of the host machinery that is relevant for the interaction/defence/symbiosis with bacteria. The authors investigate the function and biological impact of a rhizobial effector that interacts with and modifies, and curiously is modified by, legume receptors essential for symbiosis. The molecular analysis revealed a bacterial effector that cleaves a plant symbiosis signaling receptor to inhibit signaling and the host counterplay by phosphorylation via a receptor kinase. These findings have potential implications beyond bacterial interactions with plants.Bao and colleagues investigated how rhizobial effector proteins can regulate the legume root nodule symbiosis. A rhizobial effector is described to directly modify symbiosis-related signaling proteins, altering the outcome of the symbiosis. Overall, the paper presents findings that will have a wide appeal beyond its primary field.Out of 15 identified effectors from Sinorhizobium fredii, they focus on the effector NopT, which exhibits proteolytic activity and may therefore cleave specific target proteins of the host plant. They focus on two Nod factor receptors of the legume Lotus japonicus, NFR1 and NFR5, both of which were previously found to be essential for the perception of rhizobial nod factor, and the induction of symbiotic responses such as bacterial infection thread formation in root hairs and root nodule development (Madsen et al., 2003, Nature; Tirichine et al., 2003; Nature). The authors present evidence for an interaction of NopT with NFR1 and NFR5. The paper aims to characterize the biochemical and functional consequences of these interactions and the phenotype that arises when the effector is mutated.Evidence is presented that in vitro NopT can cleave NFR5 at its juxtamembrane region. NFR5 appears also to be cleaved in vivo. and NFR1 appears to inhibit the proteolytic activity of NopT by phosphorylating NopT. When NFR5 and NFR1 are ectopically over-expressed in leaves of the non-legume Nicotiana benthamiana, they induce cell death (Madsen et al., 2011, Plant Journal). Bao et al., found that this cell death response is inhibited by the coexpression of nopT. Mutation of nopT alters the outcome of rhizobial infection in L. japonicus. These conclusions are well supported by the data.The authors present evidence supporting the interaction of NopT with NFR1 and NFR5. In particular, there is solid support for cleavage of NFR5 by NopT (Figure 3) and the identification of NopT phosphorylation sites that inhibit its proteolytic activity (Figure 4C). Cleavage of NFR5 upon expression in N. benthamiana (Figure 3A) requires appropriate controls (inactive mutant versions) that have been provided, since Agrobacterium as a closely rhizobia-related bacterium might increase defense related proteolytic activity in the plant host cells.

We appreciate your recognition of the importance of appropriate controls in our experimental design. In response to your comments, we revised our manuscript to ensure that the figures and legends provide a clear description of the controls used. We also included a more detailed description of our experimental design at several places. In particular, we have highlighted the use of the protease-dead version of NopT as a control (NopT^C93S^). Therefore, NFR5-GFP cleavage in *N. benthamiana* clearly depended on protease activity of NopT and not on *Agrobacterium* (Fig. 3A). In the revised text, we carefully revied the conclusion and do not conclude at this stage that NopT proteolyzes NFR5. However, our subsequent experiments, including in vitro experiments, clearly show that NopT is able to proteolyze NFR5.

Key results from N. benthamiana appear consistent with data from recombinant protein expression in bacteria. For the analysis in the host legume L. japonicus transgenic hairy roots were included. To demonstrate that the cleavage of NFR5 occurs during the interaction in plant cells the authors build largely on western blots. Regardless of whether Nicotiana leaf cells or Lotus root cells are used as the test platform, the Western blots indicate that only a small proportion of NFR5 is cleaved when co-expressed with nopT, and most of the NFR5 persists in its full-length form (Figures 3A-D). It is not quite clear how the authors explain the loss of NFR5 function (loss of cell death, impact on symbiosis), as a vast excess of the tested target remains intact. It is also not clear why a large proportion of NFR5 is unaffected by the proteolytic activity of NopT. This is particularly interesting in Nicotiana in the absence of Nod factor that could trigger NFR1 kinase activity.

Thank you for your comments regarding the cleavage of NFR5 by NopT and its functional implications. We acknowledge that our immunoblots indicate only a relatively small proportion of the NFR5 cleavage product. Possible explanations could be as follows:

(1) The presence of full-length NFR5 does not preclude a significant impact of NopT on function of NFR5, as NopT is able to interact with NFR5. In other words, the NopT-NFR5 and NopT-NFR1 interactions at the plasma membrane might influence the function of the NFR1/NFR5 receptor without proteolytic cleavage of NFR5. In fact, protease-dead NopT^C93S^ expressed in NGR234*ΔnopT* showed certain effects in *L. japonicus* (less infection foci were formed compared to NGR234*ΔnopT* Fig. 5E). In this context, it is worth mentioning that the non-acylated NopT^C93S^ (Fig. 1B) and NopT_USDA257_ (Fig. 6B) proteins were unable to suppress NFR1/NFR5-induced cell death in *N. benthamina*, but this could be explained by the lack of acylation and altered subcellular localization.

(2) In the cleavage assay, only small portion of NFR5 could be detected for cleavage by NopT. However, this cleavage might be sufficient to suppress signaling pathways, leading to the observed phenotypic changes (loss of cell death in *N. benthamiana*; altered infection in *L. japonicus*). We do believe this is a great point, therefore, we carefully revised the conclusion about this point. Throughout the paper, we stated that the cleavage of NFR5 suppresses symbiotic signaling but not disrupt the symbiotic signaling. We also removed the conclusion that cleavage of NFR5 by NopT results in the function loss of NFR5.

(3) *N. benthamiana* co-expressing NFR1/NFR5 leads to strong cell death, which suggest that the NFR1 kinase activity might be constitutively active even in the absence of Nod factors. But why co-expression of symbiotic receptor leads to cell death and how kinase activity is active in the absence of Nod factor are not clear, which is of great interest to be studied.

(4) The proteolytic activity of NopT may be reduced by the interaction of NopT with other proteins such as NFR1, which phosphorylates NopT and inactivates its protease activity.

In our revised manuscript version, we provide now quantitative data for the efficiency of NFR5 cleavage by NopT in different expression systems used (Figure 3 and Supplemental Fig. 16). We have also improved our Discussion in this context.

Comments on latest version:The presentation of the figures and the language has greatly improved and the specific mistakes pointed out in the last review have been corrected. I especially appreciate the new images used to illustrate the observed mutant phenotypes, which are much clearer and easier to understand. The pictures used to illustrate the mutant phenotypes seem to be of more comparable root regions than before. Overall, the requested changes have been implemented, with some exceptions described below.• Figure 1: New representative images are shown for BAX1 and CERK1. These pictures are more consistent with the phenotype seen in other treatments, but since the data has not changed, I presume the data from leaf discs (where the leaf discs for these treatments looked very different) previously shown is still included. The criteria for what was considered cell death is in my opinion still not described in the legend. The cell death/total ratio has been added for all leaf discs, as requested.

Thank you so much for carefully pointing out this. Cell death in leaf disc results in the formation of necrotic plaques, which restrains pathogens within deceased cells. These plaques commonly manifest as leaf dehydration, frequently accompanied by a translucent appearance. Brown and shriveled leaf discs serve as indicators of cell death. We have added these descriptions in the figure legend of Figure 1.

• Figure 2: the discussion of the figure now emphasizes direct protein interaction. There is still no size marker in 2D or a description of size in the figure legend, making it difficult to compare the result to Figure 3. If I understand the rebuttal comments correctly, there are other bands on the blot, including non-specific bands. This does not negate the need to include the full blot as a supplemental figure to show cleaved NFR5 as well as other bands. I do not see any other clarifications on this subject in the manuscript.

Thank you for your suggestion. In the revised manuscript, we have included the kDa range for all proteins detected in Figure.2D. The full blot of Co-IP assay was shown in Fig S2 (a new supplemental data). Yes, we detected some smaller bands after immunoblot, but we cannot give clear conclusion of what these bands are based on the current study. Interestingly, these smaller bands were immunoprecipitated by anti-FLAG beads, suggesting that these bands are some truncated peptides from NFR5.

• Figure 5: From the pictures, it is now easier to understand what is meant by "infection foci". Although there is no description in the methods of how these were distinguished from infection threads, I believe the images are clear enough.

Thank you for your helpful comment. In the revised manuscript, we have added the descriptions about this experiment in the method section and in the legend in Figure 5A.

• Figure 6: The changes in the discussion are appreciated, but panel E still misrepresents the evidence in the paper, as from the drawing it still seems that the cleaved NFR5 is somehow directly responsible for suppressing infection when this was not shown.

Thank you for your thoughtful comments. We appreciate your suggestion to the schematic model to illustrate the cleavage of NFR5 to suppressing rhizobia infection. In the revised manuscript, we have changed the model in Figure 6E.

**Reviewer #2 (Public review):**
Summary:This manuscript presents data demonstrating NopT's interaction with Nod Factor Receptors NFR1 and NFR5 and its impact on cell death inhibition and rhizobial infection. The identification of a truncated NopT variant in certain Sinorhizobium species adds an interesting dimension to the study. These data try to bridge the gaps between classical Nod-factor-dependent nodulation and T3SS NopT effector-dependent nodulation in legume-rhizobium symbiosis. Overall, the research provides interesting insights into the molecular mechanisms underlying symbiotic interactions between rhizobia and legumes.Strengths:The manuscript nicely demonstrates NopT's proteolytic cleavage of NFR5, regulated by NFR1 phosphorylation, promoting rhizobial infection in L. japonicus. Intriguingly, authors also identify a truncated NopT variant in certain Sinorhizobium species, maintaining NFR5 cleavage but lacking NFR1 interaction. These findings bridge the T3SS effector with the classical Nod-factor-dependent nodulation pathway, offering novel insights into symbiotic interactions.Weaknesses:(1) In the previous study, when transiently expressed NopT alone in Nicotiana tobacco plants, proteolytically active NopT elicited a rapid hypersensitive reaction. However, this phenotype was not observed when expressing the same NopT in Nicotiana benthamiana (Figure 1A). Conversely, cell death and a hypersensitive reaction were observed in Figure S8. This raises questions about the suitability of the exogenous expression system for studying NopT proteolysis specificity.

We appreciate your attention to these plant-specific differences. Previous studies showed that NopT expressed in tobacco (*N. tabacum*) or in specific *Arabidopsis* ecotypes (with PBS1/RPS5 genes) causes rapid cell death (Dai et al. 2008; Khan et al. 2022). Khan et al. 2022 reported recently that cell death does not occur in *N. benthamiana* unless the leaves were transformed with PBS1/RPS5 constructs. Our data shown in Fig. S17 confirm these findings. As cell death is usually associated with induction of plant protease activities, we considered *N. tabacum* and *A. thaliana* plants as not suitable for testing NFR5 cleavage by NopT. In fact, no NopT/NFR5 experiments were not performed with these plants in our study. In response to your comment, we now better describe the *N. benthamiana* expression system and cite the previous articles_._ Furthermore, we have revised the Discussion section to better emphasize effector-induced immunity in non-host plants and the negative effect of rhizobial effectors during symbiosis. Our revisions certainly provide a clearer understanding of the advantages and limitations of the *N. benthamiana* expression system.

(2) NFR5 Loss-of-function mutants do not produce nodules in the presence of rhizobia in lotus roots, and overexpression of NFR1 and NFR5 produces spontaneous nodules. In this regard, if the direct proteolysis target of NopT is NFR5, one could expect the NGR234's infection will not be very successful because of the Native NopT's specific proteolysis function of NFR5 and NFR1. Conversely, in Figure 5, authors observed the different results.

Thank you for this comment, which points out that we did not address this aspect precisely enough in the original manuscript version. We improved our manuscript and now write that *nfr1* and *nfr5* mutants do not produce nodules (Madsen et al., 2003; Radutoiu et al., 2003) and that over-expression of either *NFR1* or *NFR5* can activate NF signaling, resulting in formation of spontaneous nodules in the absence of rhizobia (Ried et al., 2014). In fact, compared to the *nopT* knockout mutant NGR234*ΔnopT*, wildtype NGR234 (with NopT) is less successful in inducing infection foci in root hairs of *L. japonicus* (Fig. 5). With respect to formation of nodule primordia, we repeated our inoculation experiments with NGR234*ΔnopT* and wildtype NGR234 and also included a *nopT* over-expressing NGR234 strain into the analysis. Our data clearly showed that nodule primordium formation was negatively affected by NopT. The new data are shown in Fig. 5 of our revised version. Our data show that NGR234 infection is not really successful, especially when NopT is over-expressed. This is consistent with our observations that NopT targets Nod factor receptors in *L. japonicus* and inhibits NF signaling (*NIN* promoter-GUS experiments). Our findings indicate that NopT might be an “Avr effector” for *L. japonicus*. However, in other host plants of NGR234, NopT possesses a symbiosis-promoting role (Dai et al. 2008; Kambara et al. 2009). Such differences could be explained by different NopT targets in different plants (in addition to Nod factor receptors), which may influence the outcome of the infection process. Indeed, our work shows that NopT can interact with various kinase-dead LysM domain receptors, suggesting a role of NopT in suppression or activation of plant immunity responses depending on the host plant. We discuss such alternative mechanisms in our revised manuscript version and emphasize the need for further investigation to elucidate the precise mechanisms underlying the observed infection phenotype and the role of NopT in modulating symbiotic signaling pathways. In this context, we would also like to mention the new figures of our manuscript which are showing (i) the efficiency of NFR5 cleavage by NopT in different expression systems (Figure 3), (ii) the interaction between NopT^C93S^ and His-SUMO-NFR5JM-GFP (Supplementary Fig. 5), and (iii) cleavage of His-SUMO-NFPJM-GFP by NopT (Supplementary Figs. S8 and S9).

(3) In Figure 6E, the model illustrates how NopT digests NFR5 to regulate rhizobia infection. However, it raises the question of whether it is reasonable for NGR234 to produce an effector that restricts its own colonization in host plants.

Thank you for mentioning this point. We are aware of the possible paradox that the broad-host-range strain NGR234 produces an effector that appears to restrict its infection of host plants. As mentioned in our answer to the previous comment, NopT could have additional functions beyond the regulation of Nod factor signaling. In our revised manuscript version, we have modified our text as follows:

(1) We mention the potential evolutionary aspects of NopT-mediated regulation of rhizobial infection and discuss the possibility that interactions between NopT and Nod factor receptors may have evolved to fine-tune Nod factor signaling to avoid rhizobial hyperinfection in certain host legumes.

(2) We also emphasize that the presence of NopT may confer selective advantages in other host plants than *L. japonicus* due to interactions with proteins related to plant immunity. Like other effectors, NopT could suppress activation of immune responses (suppression of PTI) or cause effector-triggered immunity (ETI) responses, thereby modulating rhizobial infection and nodule formation. Interactions between NopT and proteins related to the plant immune system may represent an important evolutionary driving force for host-specific nodulation and explain why the presence of NopT in NGR234 has a negative effect on symbiosis with *L. japonicus* but a positive one with other legumes.

(4) The failure to generate stable transgenic plants expressing NopT in Lotus japonicus is surprising, considering the manuscript's claim that NopT specifically proteolyzes NFR5, a major player in the response to nodule symbiosis, without being essential for plant development.

We also thank for this comment. We have revised the Discussion section of our manuscript and discuss now our failure to generate stable transgenic *L. japonicus* plants expressing NopT. We observed that the protease activity of NopT in aerial parts of *L. japonicus* had a negative effect on plant development, whereas NopT expression in hairy roots was possible. Such differences may be explained by different NopT substrates in roots and aerial parts of the plant. In this context, we also discuss our finding that NopT not only cleaves NFR5 but is also able to proteolyze other proteins of *L. japonicus* such as LjLYS11, suggesting that NopT not only suppresses Nod factor signaling, but may also interfere with signal transduction pathways related to plant immunity. We speculate that, depending on the host legume species, NopT could suppress PTI or induce ETI, thereby modulating rhizobial infection and nodule formation.

Comments on revised version:This version has effectively addressed most of my concerns. However, one key issue remains unresolved regarding the mechanism of NopT in regulating nodule symbiosis. Specifically, the explanation of how NopT catabolizes NFR5 to regulate symbiosis is still not convincing within the current framework of plant-microbe interaction, where plants are understood to genetically control rhizobial colonization.While alternative regulatory mechanisms in plant-microbe interactions are plausible, the notion that the NRG234-secreted effector NopT could reduce its own infection by either suppressing plant immunity or degrading the symbiosis receptor remains unsubstantiated. I believe further revisions are needed in the discussion section to more clearly address and clarify these findings and any lingering uncertainties.

We appreciate your positive comments on the reason why NopT catabolizes NFR5 to regulate symbiosis. NopT belongs to pathogen effecftors YopT family and also cleavage *Arabidopsis* AtLYK5 and *L. japonicus* LjLYS11 which trigger immunity responses in plants. NFR5, AtLYK5 and LjLYS11 has the conserved amino acid motif at the juxtamembrane domain, leading to cleaving NFR5 by NopT during symbiosis. Besides, in plant-microbe interaction, effector HopB1 cleaves immune co-receptor BAK1 at the kinase domain to inhibit plant defense. The effect on cleavage of receptor may be positive or negative. NopT suppressing symbiosis may avoid preventing hyperinfection in the specific interaction between rhizobia and legumes. In the revised manuscript, we have emphasized this point more clearly in why NopT could reduce its own infection by either suppressing plant immunity in discussion.

**Recommendations for the authors:**

**Reviewer #1 (Recommendations for the authors):**
Evaluation of the author's responses to the reviewer comments during the first review roundReviewer's Comment:Regardless of whether Nicotiana leaf cells or Lotus root cells are used as the test platform, the Western blots indicate that only a small proportion of NFR5 is cleaved when co-expressed with NopT, and most of the NFR5 persists in its full-length form (Figures 3A-D). It is not quite clear how the authors explain the loss of NFR5 function (loss of cell death, impact on symbiosis), as a vast excess of the tested target remains intact. It is also not clear why a large proportion of NFR5 is unaffected by the proteolytic activity of NopT. This is particularly interesting in Nicotiana in the absence of Nod factor that could trigger NFR1 kinase activity.Summary of response:• NopT could be interfering with the NFR1/NFR5 complex without proteolytic cleavage• The cleaved fraction may still be sufficient to disrupt signaling pathways• Elevated abundance of NFR5 relative to WT levels• Add quantitative data for efficiency of NFR5 cleavage in different systemsEvaluation of response:• The quantification of NFR5 cleavage efficiency is welcome, and there is some discussion of the possible reasons for the large proportion of uncleaved NFR5. It is clear that there is a large difference in cleavage efficiency between L. japonicus roots and N. benthamiana.• The data is shown as a bar plot. Given that only 3 biological replicates are used, the data points should be shown, and there is too little data to provide sensible error bars. It would be better to simply make a dot-plot and indicate the mean for each sample. However, the main aim of the comment is addressed.

Thank you for your constructive comments regarding Figure S16. In the revised manuscript, we have presented these data into dot-Plot format.

Reviewer's Comment:It is also difficult to evaluate how the ratios of cleaved and full-length protein change when different versions of NopT are present without a quantification of band strengths normalized to loading controls (Figure 3C, 3D, 3F). The same is true for the blots supporting NFR1 phosphorylation of NopT (Figure 4A).Summary of response:• Quantified proportion of cleaved and full length NFR5 in different systems (S14)• Band strengths of immunoblots quantified (4B)Evaluation of response:• The quantification has been performed as requested and the data is shown as bar plots. This type of data is frequently displayed as part of the blot figure itself, printed under each respective lane, making it easier for the reader to connect the ratios to the band sizes. If data is shown in a plot, the data points should be shown on the plot, as described above.

Thank you for your constructive comments regarding Figure 3. In the revised manuscript, we have added the cleavage efficiency in the 3A-3D.

Reviewer's Comment:Nodule primordia and infection threads are still formed when L. japonicus plants are inoculated with ∆nopT mutant bacteria, but it is not clear if these primordia are infected or develop into fully functional nodules (Figure 5). A quantification of the ratio of infected and non-infected nodules and primordia would reveal whether NopT is only active at the transition from infection focus to thread or perhaps also later in the bacterial infection process of the developing root nodule.Summary of response:• Additional experiments with NGR234 or NGR234*ΔnopT* mutants find no non-infected nodules (fig. 5)Evaluation of response:• The requested quantification has been done, although the support for the findings would be stronger if also mature nodules per plant were quantified and plotted. If non-infected nodules were neither present in NGR234 or NGR234*ΔnopT*, it would still be advisable to include images of cross-sections of the fully-developed nodules.

We appreciate your positive comments on the cross-sections of the fully-developed nodules. In the revised manuscript, we have added the cross-section images of nodules in the Figure S12.